# Projecting the long-term effects of the COVID-19 pandemic on U.S. population structure

Andrea M. Tilstra [1,2] ✉, Antonino Polizzi[1,2], Sander Wagner[1] & Evelina T. Akimova [1]

The immediate, direct effects of the COVID-19 pandemic on the United States population are substantial. Millions of people were affected by the pandemic: many died, others did not give birth, and still others could not migrate. Research that has examined these individual phenomena is important, but fragmented. The disruption of mortality, fertility, and migration jointly affected U.S. population counts and, consequently, future population structure. We use data from the United Nations World Population Prospects and the cohort component projection method to isolate the effect of the pandemic on U.S. population estimates until 2060. If the pandemic had not occurred, we project that the population of the U.S. would have 2.1 million (0.63%) more people in 2025, and 1.7 million (0.44%) more people in 2060. Pandemic-induced migration changes are projected to have a larger long-term effect on future population size than mortality, despite comparable short-term effects.

While the impact of the COVID-19 pandemic has been most tragically evident in the number of deaths it caused, it has also affected fertility and immigration. From the onset of the COVID-19 pandemic through December 2022, 1.07 million deaths from severe acute respiratory syndrome coronavirus 2 (SARS-CoV-2) were recorded in the United States (U.S.). The number of excess deaths is estimated even higher, at 1.23 million[1]. This has contributed to a dramatic decline of 2.7 years in U.S. life expectancy at birth between 2019 and 2021[2]. Emerging research suggests that U.S. fertility and international migration to and from the U.S. also changed in significant ways during the COVID-19 pandemic. The total fertility rate declined by 4% between 2019 and 2020, from 1.71 to 1.64, and then increased to 1.66 in 2021[3–5]. Net migration between July 2020 and June 2021 was estimated to be 376,000, nearly half of what it was between 2019 and 2020, and 70% lower than the highest estimate of the preceding decade – 1,236,000 between 2015 and 2016[6,7].

However, our current knowledge is fragmented. There is ample evidence that the COVID-19 pandemic has disrupted mortality, fertility, and migration patterns independently. Yet, less is known about how the disruption of these processes jointly affected population counts and, consequently, future population structure (see González Leonardo and Spijker 2022[8] and Wilson et al. 2022[9] for two exceptions, in Spain and Australia). We use population projection methods to show how the unexpected and large shock of the COVID-19 pandemic is projected to continue to affect the size and distribution of the U.S. population (by age and sex) for decades to come. If not reversed, the rippling effects of the COVID-19 pandemic for the size and age distribution of the U.S. population are projected to continue to be observed because of the COVID-19 pandemic-era disruptions to all three demographic processes: mortality, fertility, and migration, and the interdependent consequences therein.

The first two years of the COVID-19 pandemic saw extraordinary increases in mortality in the U.S. Deaths were driven by those directly attributable to COVID-19 along with increases in excess mortality for other causes of death (e.g., related to drugs[10]). As in previous years, deaths were socially patterned, with individuals from historically disadvantaged backgrounds – e.g., non-Hispanic Black and Hispanic people – experiencing higher mortality rates[11–14].

While mortality rates were climbing in the U.S., fertility rates were lower in 2020 than in 2019[15]. This is likely attributable to increased

[1]Leverhulme Centre for Demographic Science, Nuffield Department of Population Health, Nuffield College, University of Oxford, Oxford OX1 1JD, UK. [2]Department of Sociology, University of Oxford, Oxford OX1 1JD, UK. ✉e-mail: andrea.tilstra@demography.ox.ac.uk

economic insecurity, leading to a decreased desire for pregnancy[16,17]. However, emerging evidence suggests that the 'baby bust' seen in 2020 was followed by a smaller "baby bump" in 2021, though this increase in fertility was largely concentrated among US-born mothers[3,18].

During the COVID-19 pandemic, migration into the U.S. was severely restricted. In March 2020, Title 42, a scantly used title of the Public Health Service Act, was enacted. Title 42, first created in 1944, authorizes the director of the Centers for Disease Control and Prevention to prohibit entry to the U.S. for individuals coming from a country where a communicable disease has been identified[19–22]. This public health provision enabled the U.S. administration to heavily restrict migration, and thus Title 42 effectively halted much of the migration to the U.S. The use of Title 42 is highly contested, though even after the U.S. Supreme Court voted to keep it in place in December 2022, it was eventually repealed in May 2023. While it remains difficult to measure net migration in the U.S., estimates from the Census Bureau suggest a 50% reduction in net migration between July 2020 and June 2021, an estimate that is nearly uniformly distributed across migrant types – immigrant visas, work visas, student visas, and refugees and asylum seekers[6] – followed by a return to pre-COVID-19 pandemic levels, and possibly higher, in 2022[7,23]. It is without a doubt that the COVID-19 pandemic, both by triggering enforcement of Title 42 and the subsequent strict travel policies into the U.S., and by decreasing the desire and opportunity for cross-national mobility contributed to lower rates of net migration during the years 2020 and 2021.

Although COVID-19 pandemic-related changes in mortality, fertility, and migration arguably affected population size directly at older, younger, and working ages, respectively, these changes may have simultaneously exerted indirect effects on the size of other age groups. For example, fewer migrants and increased mortality at reproductive ages result in fewer people available to give birth. When this is combined with slightly lower fertility rates, as was the case in the first year of the COVID-19 pandemic, the number of people born in a given year will be lower. Population projection models are commonly used to account for these direct and indirect effects on population age structure and are a useful tool for estimating future healthcare needs, childcare and housing demands, as well as economic growth, public debt, and tax revenue. Projection models are therefore a crucial instrument for economic planning and development[24,25]. The young- (<15) and old- (65+) age dependency ratios further capture the proportion of the non-working age population relative to the individuals aged 15–64, and are oft-employed, succinct measures that convey the impact of population change on future economic measures[26]. Generally, high old-age and low young-age dependency ratios are associated with concerns about rising social security costs and lower economic growth in the near and distant future, respectively.

Applying the cohort component projection model to data from the United Nations World Population Prospects (UNWPP)[27], we compare a 'baseline' projection of the U.S. population (with COVID-19) to a 'counterfactual' projection (without COVID-19) to quantify the effect that changes brought about by the COVID-19 pandemic are projected to continue to exert on the size of different population age groups and the dependency ratio until 2060. We emphasize three important findings from our projection: (1) declines in migration during the COVID-19 pandemic are projected to have the biggest long-term impact on the size of the U.S. population, (2) there are fewer projected reproductive-aged (15–49 years old) people in the U.S. for the coming four decades, and (3) increases in mortality explain the COVID-19 pandemic's long-term effects on the dependency ratio, which, while relatively small in magnitude, are expected to be visible until the late 2040s.

## Results

In the absence of the COVID-19 pandemic, we estimate that over 2.1 million (0.63%) more people would be alive in the U.S. in 2025 than currently forecasted by UNWPP. Despite gradual reductions in magnitude, this difference persists over the long-term: had the COVID-19 pandemic not occurred, we project that the population would have 1.7 million (0.44%) additional people in 2060. We see how the projected missing population is distributed by age groups and sex in Fig. 1, which shows the absolute (panel A) and relative (panel B) difference in population size between the baseline and counterfactual projections by year. The largest absolute short-term population reduction happens among 15- to 49-year-olds and is greater in magnitude for males than females. Over time, the population loss in this age group is projected to become less pronounced but will likely stay at about 580 thousand fewer people in 2060, combined for males and females. In contrast, the largest relative short-term reduction is observable in the population 85 and older (almost 3.5% in males and 2.0% in females). Over time, the missing population relative to baseline is projected to decrease to a level of around 0.4% in 2060 for both males and females in all age groups. Across all age groups under 85, we project a cyclical increase, then decrease, followed by a later increase in the difference between baseline and counterfactual projections, suggesting a reverberation of the COVID-19 pandemic over time, even after rates of mortality, fertility, and migration return to their pre-COVID-19 pandemic trajectories.

The notable restructuring of the U.S. population is best exhibited in Fig. 2, which shows the missing population by age and sex (i.e., baseline minus counterfactual), in absolute (panel A) and relative (panel B) terms for three years: 2025 (left), 2040 (middle), and 2060 (right). These panels are available at https://doi.org/10.17605/osf.io/te592 in animated form for the years 2020–2060. The panels illustrate the persistence of the changes in population structure due to the COVID-19 pandemic. Not only do the first order effects of COVID-19 pandemic-induced population changes persist in the population pyramid over decades, but we also observe second order effects brought about by changes in the reproductive-age population during the early years of the COVID-19 pandemic. These decreases in the population's childbearing potential result in fewer births in the COVID-19 pandemic years (birth cohorts 2020–2024) and in the years after the COVID-19 pandemic (birth cohorts 2025 and later) than would have been predicted without the COVID-19 pandemic. One prominent change happens with the 2020 cohort, where there are over 42 thousand (2.34%) fewer females and 44 thousand (2.33%) fewer males in 2025 than would have been expected in the absence of the COVID-19 pandemic. As already highlighted in Fig. 1, the absolute effects are slightly larger in the middle age brackets in 2025, whereas relative effects clearly dominate in the older age brackets.

Although it is unlikely that mortality, fertility, or migration were entirely unaffected by COVID-19 pandemic changes in the respective remaining demographic processes, we attempt to isolate the independent contributions from mortality, fertility, and migration on population structure by running our model as though counterfactual conditions only applied to one demographic process at a time (see Methods). Figure 3 highlights this for the year 2040, roughly the midpoint of our study period, showing the absolute (panel A) and relative (panel B) difference in population size if the COVID-19 pandemic had not affected mortality (left) fertility (middle), or migration (right). As with Fig. 2, these panels are available at https://doi.org/10.17605/osf.io/te592 in animated form for the years 2020–2060. These panels confirm two known facts about the COVID-19 pandemic. First, that in the U.S., mortality was most disrupted for older-aged individuals, and more so for men than for women. Second, during 2020 the U.S. total fertility rate was lower than in previous years, resulting in an estimated 35 thousand (1.87%) fewer females and 37 thousand (1.88%) fewer males in that birth cohort in 2040. Finally, this figure offers valuable new information about the magnitude of the impact that migration changes had in the U.S. The halting of in-migration under Title 42 likely contributed to a reduction in working-aged individuals, of which the biggest reduction was among reproductive-aged individuals (660 thousand people; 0.41%) and a smaller reduction (220 thousand

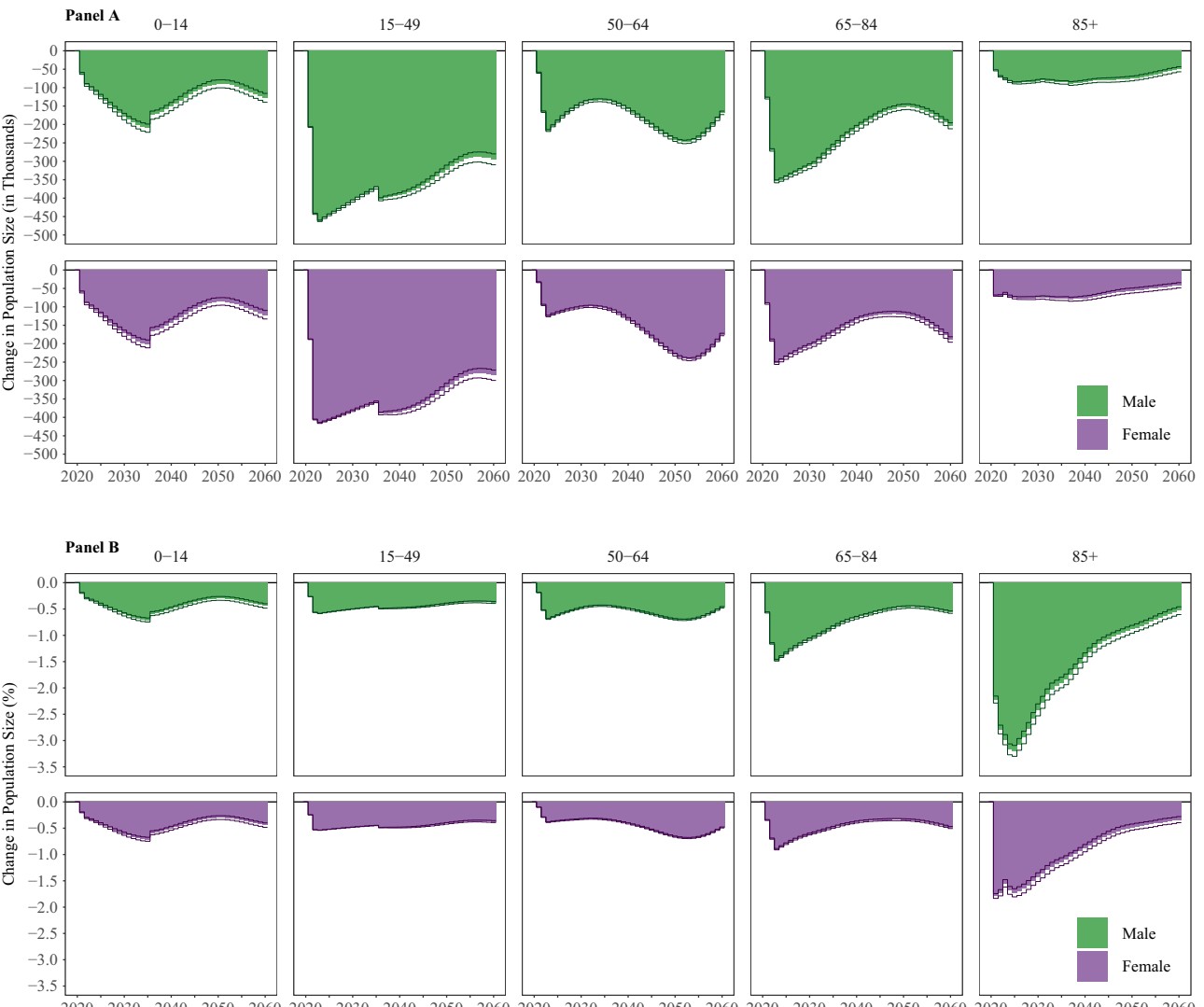

**Fig. 1 | Difference in Population Size, by Age Group and Sex.** Absolute (A) and Relative (B) Differences in Population Size by Age Group and Sex between Baseline (with COVID-19) and Counterfactual (without COVID-19), 2020–2060. *Notes. Data* *come from the United Nations World Population Prospects and the authors' own* *estimates. Lines represent 95% confidence intervals and bars represent the median,* *estimated from n = 1000 stochastic projections.*

people; 0.34%) was among individuals aged 50–64. The reverberation of this population loss is expected to be felt long into the future. Fewer people at reproductive ages means fewer births, thus later cohorts are subsequently smaller; this is a cycle that may carry on for many years.

The smaller-than-anticipated population in all age groups indeed has consequences for dependency ratios – the size of the non-working-age population (younger than age 15 and older than age 64) as a percentage of the working-age population (ages 15 to 64). We estimate that, in the aftermath of the COVID-19 pandemic, the overall dependency ratio will be 0.16 percentage points lower in 2025 than it would have been in the absence of the COVID-19 pandemic (Fig. 4, panel C). A smaller dependency ratio means that there are fewer dependents per working-aged person. When we examine this pattern for each demographic process independently, we see that the initial negative effect of the COVID-19 pandemic on the dependency ratio was driven by changes in mortality. However, this mortality effect was to some extent offset by changes in migration, which increased the dependency ratio. Changes in fertility during the COVID-19 pandemic had only limited effects on the overall dependency ratio. Over time, the negative effect of the COVID-19 pandemic on the overall dependency ratio is expected to slowly decrease and turn slightly positive between 2040 and 2060. Figure 4, panel B, shows that the lower dependency ratio between 2025

and 2040 (the first two red dots) is likely driven by the negative impact of mortality on the old-age dependency ratio, i.e., the ratio of individuals aged 65 and older to individuals aged 15–64. In contrast, Fig. 4, panels A and B, show that the higher dependency ratio between 2040 and 2060 (the last two red dots) can be explained by the positive effect of migration on the young-age and old-age dependency ratios (see also Fig. 3).

Figure 5 further disaggregates the COVID-19 pandemic's effects on the population age structure by showing the change in the proportion of the population in each age group that is attributable to the COVID-19 pandemic's effects on the different demographic components. This figure reiterates the previous point that, whereas the COVID-19 pandemic's effect on mortality led to a rise in the share of the population at younger ages and a decline in the share of the population at older ages, the COVID-19 pandemic's effect on migration operated in the opposite way. While the separate and joint effects of the three demographic processes on population structure 'fade out' with time, the small effect sizes present in the early years after the COVID-19 pandemic are noteworthy. For example, due to the COVID-19 pandemic's effects on mortality, fertility, and migration (panel 'All'), the share of the population located in the age groups 0–14 and 15–49 is projected to be around 0.05 percentage points larger in 2025,

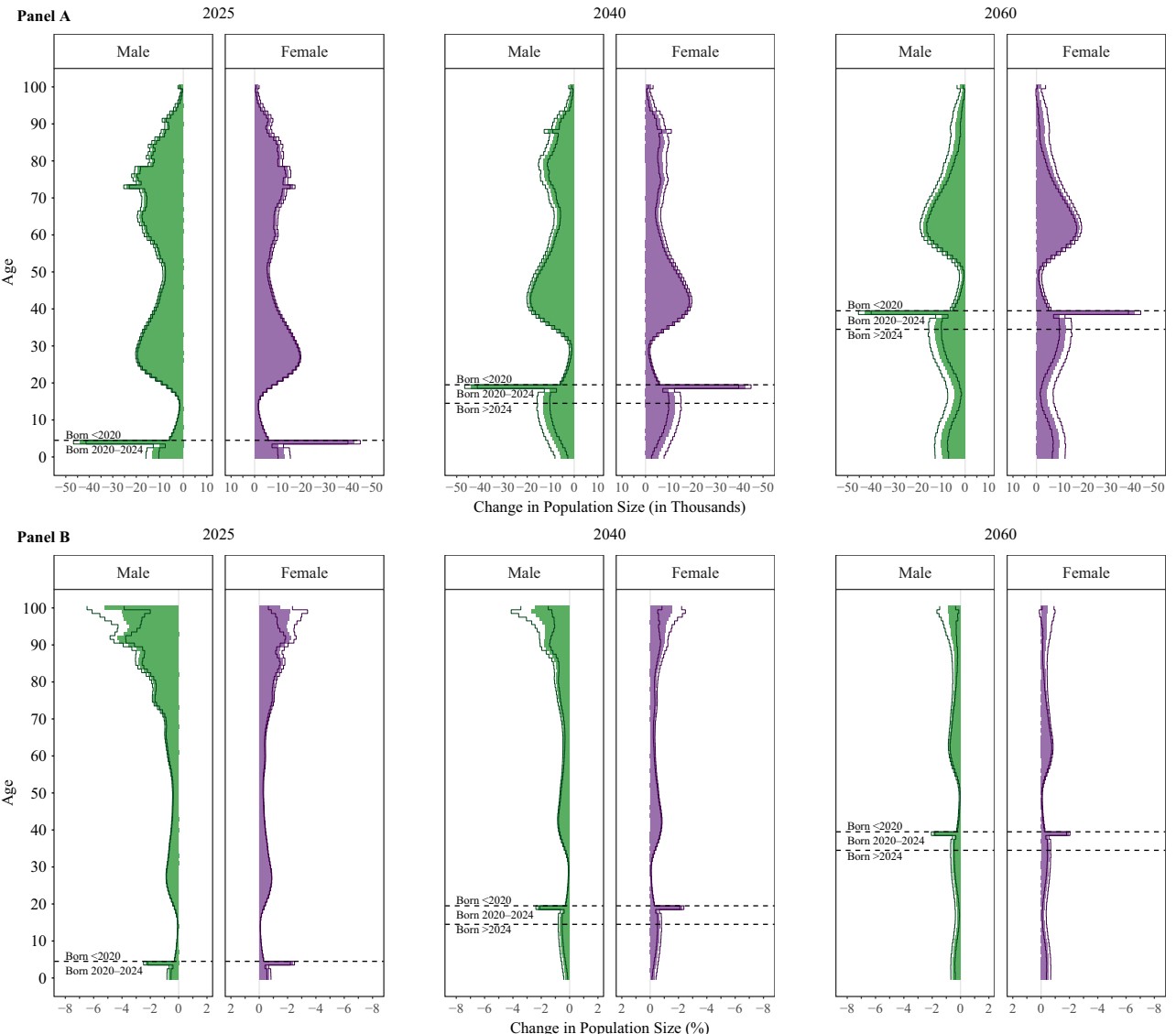

**Fig. 2 | Difference in Population Size, by Year.** Absolute (A) and Relative (B) Difference in Population Size by Age and Sex for Years 2025 (left), 2040 (middle), 2060 (right). *Notes. Data come from the United Nations World Population Prospects and the authors' own estimates. Lines represent 95% confidence intervals and bars represent the median, estimated from n = 1000 stochastic projections.*

indicating that the COVID-19 pandemic only led to a minor reshuffling in the age composition of the population.

## Discussion

Our study provides first results on how the COVID-19 pandemic's reshaping of the U.S. population is expected to repercuss into the future. Despite the general perception that the COVID-19 pandemic mainly affected old populations, our projections show that population pyramids will exhibit consequences of the pandemic until at least 2060. These rippling effects are expected when modeling the consequences of COVID-19 pandemic-induced changes in all three processes: mortality, fertility, and migration. We highlight three of the most important results from our study.

First, among the three demographic processes, the loss of net migration during the COVID-19 pandemic years is expected to have the biggest long-term impact on the size of the U.S. population. In light of concerns about below-replacement fertility and baby boomer cohorts reaching retirement age, migration represents one important mechanism for slowing down population aging. The number of resettled people in the U.S. has been declining since 1980 but declined even more dramatically after the Trump administration's 2017 Executive Order titled "Protecting the Nation from Foreign Terrorist Entry into the United States"[6,28]. Then, after the enactment of Title 42 in March 2020, immigration and resettlement to the U.S. reached the lowest level of the past forty years. Title 42 was harmful for hundreds of thousands of people and ultimately resulted in the expulsion of over 1 million migrants and asylum seekers at the U.S. border, a decision that had no clear statistical relationship with reducing COVID-19 cases[19,29]. Our results show that the decline in migration resulted in the loss of U.S. population at all ages, but especially at working and reproductive ages. This result highlights that the COVID-19 pandemic's effect on migration is more consequential for population size than its effect on mortality, a finding that is consistent with a similar study on Spain[8]. Government policy responses during crises can have profound effects on the population, through entirely different channels than their desired effect.

Second, in the next four decades there is projected to be fewer reproductive-aged (15–49 years old) people in the U.S. This is a result of fewer migrants in childbearing ages, as well as, to a lesser extent, COVID-19 pandemic deaths and second-order implications of

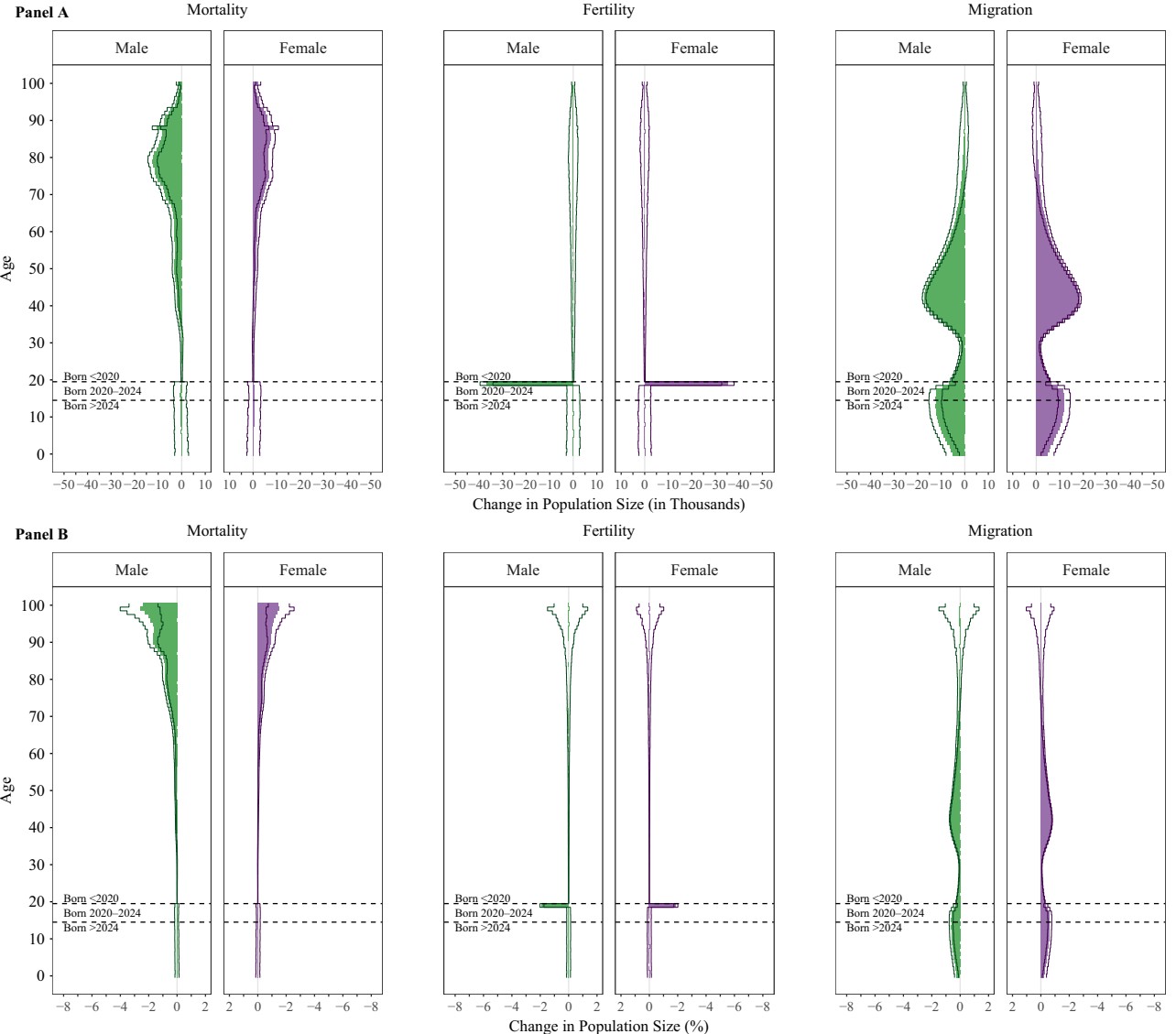

**Fig. 3 | Difference in Population Size, by Demographic Factor.** Absolute (A) and Relative (B) Difference in Population Size by Age and Sex with only Mortality Change (left), only Fertility Change (middle), and only Migration Change (right), 2040. *Notes. Data come from the United Nations World Population Prospects and the authors' own estimates. Lines represent 95% confidence intervals and bars represent the median, estimated from n = 1000 stochastic projections.*

migration and mortality for never-born children. Our estimates are likely conservative, as the effects of long COVID, or the prevalence of COVID-19 symptoms long after infection, remain to be seen. Long COVID is similar to other post-acute infections in its ability to cause health complications and disabilities[30,31]. While less is known about its mortality consequences, it stands to reason that long COVID will be a future contributor to premature deaths.

Third, the high mortality rates of the older age population during the COVID-19 pandemic have led to a small reduction in the U.S. dependency ratio. The magnitude of this reduction is attenuated by missing migration, which by itself would likely increase the dependency ratio. In 2025, almost one half of the reduction in the dependency ratio due to mortality is projected to be balanced out due to missing migration. The balance between population health and national economic stability remains a point of discussion in the U.S.[32-34]. The economic stimulus for COVID-19 pandemic relief and public health policies were important for alleviating the individual economic burden brought on by the COVID-19 pandemic and for aiding in the reduction of COVID-19 cases and mortality, but also placed extraordinary fiscal burden on the U.S. Our dependency ratio

projections provide indicators for how demographic changes brought about by the COVID-19 pandemic might continue to affect public finances in the long-term. It should also be noted that, while the dependency ratio is projected to remain slightly smaller as an effect of the COVID-19 pandemic, dependence on working-age individuals may increase due to higher healthcare needs among the older population following the COVID-19 pandemic. Additionally, we note that our calculations of dependency ratio are relatively simplistic. More nuanced calculations of dependency ratios (e.g., the "non-working-aged" dependency ratio) necessitate estimations of the number of working vs. non-working people at each age[35-37], and this data are not available in projected form from the UNWPP.

Although the UNWPP data represent a gold standard in terms of population projections, our counterfactual analysis is subject to three limitations. First, our findings are based on UNWPP's medium scenario, i.e., not the most aggressive or the most conservative estimate. As the baseline mortality, fertility, and migration rates and counts represent forecasts themselves, they are subject to uncertainty, which is carried over to our counterfactual estimates. We attempt to mitigate this by focusing on the difference between baseline and counterfactual

### Change in Dependency Ratio

| | 2025 | 2040 | 2060 | Trend |
|---|---|---|---|---|
| **Panel A: Young-age** | | | | |
| All | 0.03 | −0.01 | −0.01 | |
| Mortality | 0.04 | 0.01 | 0.00 | |
| Fertility | −0.03 | 0.01 | −0.01 | |
| Migration | 0.02 | −0.02 | 0.00 | |
| **Panel B: Old-age** | | | | |
| All | −0.19 | −0.05 | −0.04 | |
| Mortality | −0.28 | −0.18 | −0.05 | −0.28 |
| Fertility | 0.00 | 0.01 | 0.01 | |
| Migration | 0.09 | 0.11 | 0.00 | |
| **Panel C: Total** | | | | |
| All | −0.16 | −0.06 | −0.05 | |
| Mortality | −0.24 | −0.17 | −0.05 | |
| Fertility | −0.03 | 0.02 | 0.00 | |
| Migration | 0.11 | 0.09 | 0.00 | 0.14 |

**Fig. 4 | Change in Dependency Ratio.** Change in Dependency Ratios for Young-age (A, top), Old-age (B, middle), and Total (C, bottom). *Notes. Data come from the United Nations World Population Prospects and the authors' own estimates. Dependency ratios are calculated as the percentage of non-working population (<15 and > 64) to working-aged persons for each component separately and in total. Trendlines show 2020–2060, with dots at years 2025, 2040, 2060. Dashed line is equivalent to no change. Young-age = <15 / (15–64); Old-age = ≥65 / (15–64); Total = (<15 + ≥ 65) / (15–64). Estimates for "mortality" indicate the projected difference in baseline vs. counterfactual dependency ratio if only mortality had not changed during the pandemic, "fertility" if only fertility had not changed, and "migration" if only migration had not changed. Lines, dots, and reported values represent the median, estimated from n = 1000 stochastic projections.*

scenarios. Thus, because mortality, fertility, and migration conditions are set to equal after 2024, there is little room for forecasting errors to compound over time, as these will mostly cancel out. Moreover, the published UNWPP forecasts for the year 2022 correspond well with preliminary estimates of mortality, fertility, and migration[4,6,18], generating further trust in our baseline and counterfactual estimates for the COVID-19 pandemic period. Additionally, due to the nature of counterfactual analyses, it is not possible to truly know what observed rates and counts would have been in the absence of the COVID-19 pandemic. While we estimate these to the best of our ability, all analyses must be considered with this limitation in mind.

Second, our finding that changes in migration during the COVID-19 pandemic are projected to exert the biggest long-term effects on population size may partially be driven by the lack of adequate age- and sex-specific migration counts for the U.S. and the application of model migration schedules[38] for both the baseline and the counterfactual scenario. We assume a family migration schedule, with migrants concentrated in young and working ages. This also means that the second-order effects of migration through never-born children are particularly large in our study. Immigration to the U.S. has traditionally been concentrated in working ages[39] and it is plausible that the largest declines in migration during the COVID-19 pandemic occurred in these age groups. Although it is entirely possible that migration decreased more in other age-groups, including ages older than reproductive ages, existing data on foreign-born immigration to the U.S. indicate that different types of migration (i.e., refugees/asylum seekers, students, work visas, immigrant visas) were similarly affected during the COVID-19 pandemic[6]. Moreover, the enactment of Title 42 during the COVID-19 pandemic contributed to declines in migration to the U.S. and targeted a broad range of countries[19–22]. Thus, our decision to use similar migration schedules for our baseline and counterfactual scenario appears justified. While we are limited by the lack of migration data at smaller temporal windows (e.g., month or week), future work with better data availability might consider analyzing this to gain a more nuanced understanding of how these processes vary across other temporal dimensions.

Third, following UNWPP, we assume that mortality, fertility, and migration return to their pre-COVID-19 pandemic trajectories after a few years. There is inconclusive evidence about what signals the end of a pandemic or epidemic[40], so it is possible that the assumptions from UNWPP are incorrect. Should that be the case, and mortality continue to remain higher than expected, and fertility and/or migration continue to remain lower than expected, then our estimates represent an underestimation. The indirect consequences of the COVID-19 pandemic may continue to negatively affect the U.S. mortality, fertility, and migration environments well into the future, and we are not able to measure these indirect consequences here. First, long COVID and unmet healthcare needs during the COVID-19 pandemic may increase the risk of mortality in the long run. Other consequences of the COVID-19 pandemic, such as the loss of next of kin[41], learning loss[42], or racist and xenophobic behavior against Asians and Asian-Americans[43,44] may also exert negative effects on population health and mortality for generations to come. Second, the experience of economic uncertainty and stress related to the balancing of work and childcare obligations during the COVID-19 pandemic may have raised doubts among some couples about having (additional) children in the future[45,46]. Finally, migration to the U.S. may remain below expected levels in the future, as some individuals who would have migrated to the U.S. may have died during the COVID-19 pandemic, or established families in their country of origin or other countries with less restrictive migration policies. Based on these reflections about the potential long arm of the COVID-19 pandemic, the findings presented in this manuscript, which assume a short pandemic shock, most likely represent a lower bound.

Despite these limitations, our approach is valuable because it considers the interacting effects of changes to population processes. The U.S. will face a variety of public health challenges in the coming years that may have long-lasting effects on the population size and structure, and the COVID-19 pandemic is just one of these challenges. The maternal health and midlife mortality crises are likely to affect the U.S. population through multiple avenues. Demographic predictions warn that a total abortion ban could lead to excess pregnancy-related deaths of nearly 25%[47,48], while other work suggests that it may have consequences for in-vitro fertilization rates, contributing to a decline in number of births[49]. Additionally, if the midlife mortality crisis in the U.S. persists[50,51], and if rising mortality rates from the opioid epidemic are not curtailed, then deaths among reproductive-aged people will

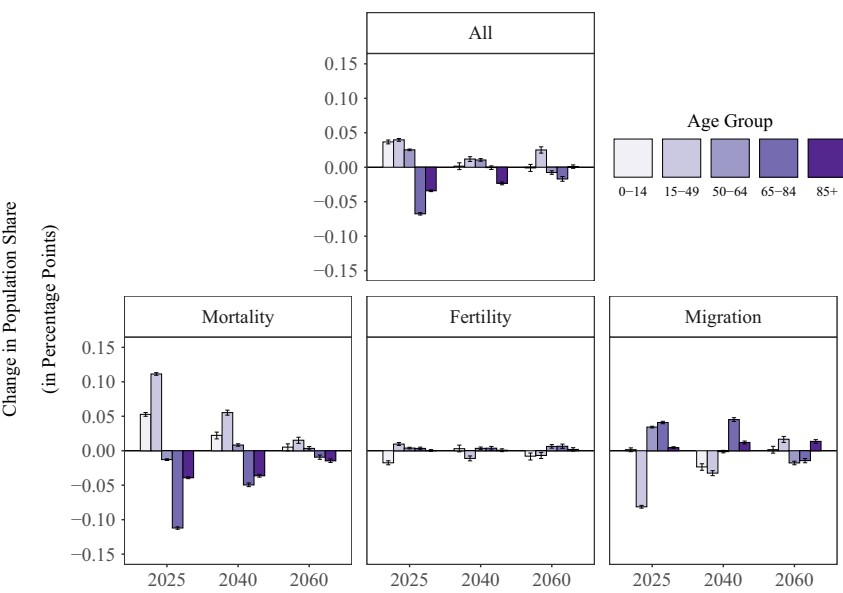

**Fig. 5 | Change in proportion of population in each age group.** *Notes. Data come from the United Nations World Population Prospects and the authors' own estimates. Whiskers represent 95% confidence intervals and bars represent the median, estimated from n = 1000 stochastic projections.*

continue to rise, resulting in fewer people at young adult and midlife ages. Applying the cohort component projection method to these crises will be valuable for understanding the magnitude of their consequences for the U.S. population. It will also be valuable to apply this approach to other countries (beyond Spain and Australia[8,9]), as the COVID-19 pandemic unequally affected each nation.

The consequences of the COVID-19 pandemic are not over. They ripple beyond immediate, independent changes to mortality, fertility, and migration to affect the population structure of the United States for decades to come. It is thus important to move from process-specific models to a broader and more informative approach that accounts for co-occurring disruptions in mortality, fertility, and migration. As this paper shows, such a design is a powerful tool for quantifying the relative size of different effects of the COVID-19 pandemic and for projecting their effects over time. Because the United States is known for having exceptionally high COVID-19 mortality[52], it is important to note that COVID-19 pandemic-induced migration changes are projected to have a comparatively large and longer-lasting effect on population size.

## Methods

The data used in this study are collected by an external source, fully anonymized and cannot be traced back to the individual from whom it came, thus exempt from ethics review by the University of Oxford Medical Sciences Interdivisional Research Ethics Committee. We use mortality rates and migration counts (by year, age, and sex), female fertility rates (by year and age), and sex ratios at birth (by year) as input in a two-sex cohort component projection model to project the population on 1st January 2020 forward in time[53]. Mortality, fertility, and migration forecasts from the 2022 United Nations World Population Prospects (UNWPP) are used to generate 'baseline' (with COVID-19) population projections by age (0–100 +) and sex for 1st January of each year in the period 2021–2060 (UN 2022)[27]. The UNWPP data for the U.S. are compiled from several federal statistical agencies – the US Census Bureau and the Vital Registration offices – in conjunction with international estimates. More detailed information on the UNWPP data is available in our Appendix and via the UNWPP methodology site.

UNWPP makes assumptions about mortality, fertility, and migration that inform our model specification. First, UNWPP expects that

baseline mortality in the near future will be the same as expected without the COVID-19 pandemic. Thus, we derive 'counterfactual' (without COVID-19) mortality rates for the years 2020–2024 by performing linear interpolation on baseline log mortality rates between 2019 and 2025 and taking the antilogarithm (by age and sex). Second, UNWPP expects no long-term effects of the COVID-19 pandemic on fertility. Thus, counterfactual age-specific fertility rates are derived only for the year 2020 by taking the average of the baseline total fertility rates in 2019 and 2021 and applying the baseline age-specific fertility pattern in 2021. Finally, UNWPP expects that U.S. net migration will return to its pre-COVID-19 pandemic trajectory in 2022. Thus, counterfactual total net migration counts for 2020 and 2021 are obtained by linearly interpolating baseline total net migration counts between 2019 and 2022. All total net migration counts are distributed across age and sex using model age- and sex-patterns of migration[38] implemented in the R package "DemoTools"[54] A "family" model of migration is assumed.

The difference between baseline and counterfactual population counts (by year, age, and sex) represents our estimated absolute annual difference in expected population size ('missing population') due to the COVID-19 pandemic. The relative difference is expressed as a percentage of the baseline population, i.e., missing population divided by baseline population and multiplied by one hundred. Throughout all analyses we use a stochastic modeling approach. For mortality, we treat each projection step as a binomial experiment, where we set the number of trials $n$ equal to the age-specific population count and the success probability $p$ equal to the survival probability in the respective age group. For fertility, we treat the number of live births in each year as a random draw from a Poisson distribution, where we set the mean equal to the number of live births that would have resulted if the age-specific fertility rates had been applied deterministically to the population. We repeat our population projections 1,000 times and present the median value for each indicator, and in all figures the thin lines or whiskers represent the 95% confidence intervals. Importantly, this method for generating confidence intervals does not address underlying uncertainty around the estimation of birth and death rates, as well as migration counts. We refer to our supplementary material for supplementary analyses with alternate assumptions. We cease analyses

in 2060 because of increasing forecasting uncertainty in the UNWPP estimates after that year.

The cohort component projection model does not allow for the decomposition of the total missing population into the individual contributions made by mortality, fertility, and migration. Thus, to estimate the contribution of each demographic process, we re-estimate the model three separate times, holding two of the three processes at their baseline values and setting only the third to counterfactual values. We calculate total dependency ratios for the baseline and counterfactual scenarios as the ratio between non-working-aged persons (younger than age 15 and older than age 64) and working-aged persons (ages 15–64). Old-age dependency ratio is the ratio of the number of persons aged 65 and older to persons aged 15–64; young-age dependency ratio is the ratio of the number of persons younger than age 15 to persons aged 15–64. All results shown are the absolute differences between baseline and counterfactual. The link to our OSF repository, for replication purposes, is available here: https://doi.org/10.17605/osf.io/te592.

### Reporting summary

Further information on research design is available in the Nature Portfolio Reporting Summary linked to this article.

## Data availability

The United Nations World Population Prospects (UNWPP) 2022 data used in this study are available, under a Creative Commons license BY 3.0 IGO. The data are publicly available at https://population.un.org/wpp/. For ease of replication, we have created a project dataset, available in our OSF repository (https://doi.org/10.17605/osf.io/te592). This project dataset contains just the measures used in our analyses. This will allow any reader or user to replicate our analyses exactly. Data used in the supplementary material come from the Congressional Budget Office, the United States Census Bureau International Database, the Human Mortality Database, the Human Fertility Database, and the National Vital Statistics System. Data from these sources is publicly available and information on access can be found in the supplementary material.

## Code availability

For the ease of replication, all R scripts and project data, including the data underlying all figures, are available at our OSF repository: https://doi.org/10.17605/osf.io/te592. R Version 4.2.2 is used in all analyses.

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

## Acknowledgements

We acknowledge funding and support from the Leverhulme Trust (Grant RC-2018-003) for the Leverhulme Centre for Demographic Science (AMT, AP, SW, ETA), the European Research Council (grant ERC-2021-CoG-101002587) (AMT, AP), the UK Research and Innovation (UKRI) under the UK government's Horizon Europe funding guarantee (EP/X027678/1) (AMT), the European Research Council ERC Advanced Grant CHRONO 835079 (ETA), the Oxford Sociology Inspiration Fund (AMT, AP), Nuffield College (AP), the Clarendon Fund (AP), and the International Max Planck Research School for Population, Health and Data Science (IMPRS-PHDS) (AP). Previous versions of this manuscript benefitted from feedback provided by the Leverhulme Centre for Demographic Science's Health Inequality Working Group and Alyson van Raalte. The content of this manuscript is solely the responsibility of the authors and does not necessarily represent the official views of the United Nations, the ERC, UKRI, the Clarendon Fund, or the Leverhulme Trust.

## Author contributions

A.M.T. and A.P. conceptualized the paper. A.P. curated data and conducted formal analyses. A.M.T. wrote the first draft. A.M.T., A.P., S.W., and E.A. contributed to the visualization, writing, and editing, and all read and approved the final version of the manuscript.

## Competing interests

The authors declare no competing interests.
