## [Peer Review File · Nature Communications]

Projecting the Long-term Effects of the COVID-19 Pandemic on U.S. Population StructureReviewers' Comments:

Reviewer #1:

Remarks to the Author:

Dear authors,

I read with great interest your article on "The Long-term Effects of the COVID-19 Pandemic on U.S. Population Structure". As the authors state, while there is ample evidence that the pandemic has disrupted mortality, fertility, and migration patterns independently, little is known about how the disruption of these processes jointly affected population counts and, consequently, future population structure.

I have a number of comments and suggestions that I hope will help to improve the quality of the paper. First I'll briefly mention my main concern before highlighting minor points:

Main concern:

How the population structure will be different in 2060 as a result of the COVID-19 pandemic is based on a lot of conjecture. The authors themselves assume that "following UNWPP ... mortality, fertility, and migration [will] return to their pre-pandemic trajectories after a few years". It is therefore unclear why the authors suppose in the same sentence that "the pandemic may continue to negatively affect the U.S. mortality, fertility, and migration environments well into the future" (lines 222-224). The first part of this statement is incongruent with the latter part. Indeed, evidence from elsewhere corroborates the temporality of the demographic consequences of the pandemic (see some examples below).

I would rather concentrate on shorter term effects of the pandemic, say until 2030 or 2040 at the most, using different possible scenarios, i.e. similar to the Spanish study the authors cited as well as another study that should have been mentioned, which produced post-covid scenarios for Australia based on possible short, moderate and long-term effects:

Wilson, T., J. Temple, and E. Charles-Edwards. 2021. Will the COVID-19 pandemic affect population ageing in Australia?, *Journal of Population Research*, 1–15. <https://doi.org/10.1007/s12546-021-09255-3>

What was missing:

In the abstract, the authors provide the main result, namely that "If the pandemic had not occurred, the population of the U.S. would have 2.1 million (0.63%) more people in 2025, and 1.7 million (0.44%) more people in 2060" yet there was no table or graph that showed a time trend (say 2010-2060) of the US population. In fact, similar to the Australian and Spanish studies it would have been helpful to have had graphs of each demographic component as well, comparing the UNWPP and counterfactual scenarios.

Minor comments:

Lines 29-30 The authors mention that the TFR "declined by 4%, from 1.71 to 1.64, between 2019 and 2020 but "seemingly recovered in 2021"". Could you state the exact TFR for 2021 rather than using a value judgement ("seemingly recovered")?

In lines 38-41 the authors state the "The rippling effects of the pandemic for the size and age-distribution of the U.S. population will continue to be observed because of the prolonged consequences of disruptions to all three demographic processes: mortality, fertility, and migration". How sure are the authors of the supposed prolonged consequences of each demographic component? In several countries, migration reached record levels in 2022, in part due to the war in Ukraine, while life expectancy has also recuperated. For instance, in the Netherlands, the total population grew more in

2022 than double than in 2019, i.e. the year prior to the start of the Covid-19 pandemic: <https://www.cbs.nl/en-gb/news/2023/01/population-growth-almost-doubled-in-2022>. Even if we disregard the influx of Ukrainians, the increase would have been greater than in 2019. Closer to the US, i.e. Canada, also saw record-high population growth in 2022 where migration was responsible for 95.9%: <https://www150.statcan.gc.ca/n1/daily-quotidien/230322/dq230322f-eng.htm> These are just 2 examples I checked, but I urge the authors to rephrase this based on what has already been observed in the US and elsewhere.

88-89 Do the authors really think that "increases in mortality dominate the pandemic's long-term effects on the dependency ratio ... will likely be visible until the late 2040s"? Please provide references from medical/public health literature to substantiate this claim.

143-144 I found the explanation that "A smaller dependency ratio means that there are fewer dependents per working-aged person, so this result can be interpreted as meaning that the pandemic contributed to fewer people in need of financial and social support" rather simplistic, basically because utilizing the old-age dependency ratio (OADR) as an exclusive gauge for financial burden in the aftermath of the pandemic undermines the consideration of other significant variables pertinent to economic productivity. While the OADR has been commonly used to measure the proportion of the elderly population to the working-age population, it fails to account for other essential factors such as changes in labor force participation, unemployment, and associated benefits that affects the denominator here. Neglecting these critical variables can lead to inaccurate estimations of the consequences of the pandemic on the economy. Other, more useful, indicators, include Pension Worker Ratio (1), the ratio of non-workers to workers irrespective of their age (2), or, even better, the 'Non-Working-aged' Dependency Ratio, which is the number of non-working persons over 60 per full-time equivalent worker (3).

1. Bongaarts J (2004) Population aging and the rising cost of public pensions. *Population and Development Review* 30 (1):1-23.
2. Vaupel JW, Loichinger E (2006) Redistributing Work in Aging Europe. *Science* 312 (5782):1911-1913.
3. Tyers R, Shi Q (2007) Demographic change and policy responses: Implications for the global economy. *The World Economy* 30 (4):537-566.

174-175 While the author's result "counterintuitively highlights that the pandemic's effect on migration is more consequential for population size than its effect on mortality", it is consistent with the Spanish study that the authors cited earlier on (ref 7).

178-180 The authors state that "in the next four decades there will likely be fewer reproductive-aged (15-49 years old) people in the U.S. This is a result of pandemic deaths and fewer migrants in childbearing age, as well as of the second-order implications this has for never-born children". Sure, it this was partly due to pandemic deaths in childbearing age but the % of mortality from 15-49 year olds was very very low. I don't know how low in the US as I couldn't find it in the OSF repository link but is probably less than 5% (based on own published research I can say it was 3% in Spain in 2020)). So, second-order implications would be really minimal.

229-231. The authors state that "The elevated pandemic mortality among population groups with traditionally high fertility, such as non-Hispanic Black and Hispanic people, and the reduced immigration of foreign-born individuals, who usually have higher fertility than native-born individuals, imply that fertility rates may remain depressed for an extended time period". This statement appears to be more on conjecture than facts. I'm not sure what the authors consider as being "high fertility" but both African American and Hispanic fertility rates have dropped quite considerably in recent decades, and are only marginally above those of White non-Hispanics (for a comparison between 2010 and 2020 see e.g. <https://www.newgeography.com/content/007528-us-total-fertility-rates-toward-europe>). Please also provide a reference for the claim that "foreign-born individuals usually have

higher fertility than native-born individuals".

Reviewer #2:

Remarks to the Author:

see attached pdf

Reviewer #3:

Remarks to the Author:

The paper is a projection exercise based on the UN WPP2022 revision. UN Population Division in its forecasts tried to incorporate available information on the consequences of COVID-19 in defining scenarios for future population trends. The authors tried to modify UN scenarios for mortality, fertility, and migration in such a way that from their point of view would eliminate the consequences of COVID-19 to measure the net effect of COVID-19 on the US population structure.

In any population projection, the resultant population composition is fully defined by scenarios of 3 components of population change – fertility, mortality, and migration. It makes sense to discuss the results if projection scenarios make sense. Unfortunately, I am not convinced of the meaningfulness of counterfactual scenarios.

I see this paper as just a pure exercise that is irrelevant to a real-world situation. I do not see any justification for counterfactual scenarios.

First of all, why are UN data taken as the baseline? These data do not necessarily reflect the real situation better than the data from the US Bureau of the Census. UN Population division is doing a great job, but they work with more than 200 countries and of course, they do not have enough men power to deal with each country in the same detail as the Bureau of Census can deal with the US. For example, the UN estimated net migration to the US in 2021 as about 560 thousand migrants. U.S. Census Bureau estimated Net international migration between 2021 and 2022 above 1 million. <https://www.census.gov/newsroom/press-releases/2022/2022-population-estimates.html> Only this fact already undermines the conclusions of the paper.

Why do the authors expect that COVID-19 had a negative effect on fertility and adjust TFR for 2021 to inflate it? It could very well be that fertility without COVID in 2021 could have been even lower than in 2019 if the TFR declining trend that began around 2007-2008 continued.

Mortality scenarios cause questions as well. CDC estimates life expectancy for males and females in 2021 as 73.2 and 79.1 years of life.

(https://www.cdc.gov/nchs/pressroom/nchs_press_releases/2022/20220831.htm) The corresponding estimates of the UN are more than 1 year higher - 74.3 and 80.2 years.

UN scenarios for life expectancy for males and females do not foresee any "harvesting" effect at least in the medium scenario used in the paper. However, it is well possible that after periods with excess mortality, there could be a period of mortality decline in subsequent times because the most fragile individuals passed away. For example, a very large number of COVID-19 deaths occurred in nursery homes. In many cases, COVID-19 just sped up the deaths that would have occurred soon anyway since life expectancy in nursery homes is very short. On the other hand, as the authors also noted no one knows about the long-term effects of COVID-19 and its effects on the humans who experienced the disease. So this is more a probabilistic issue. And that brings me to another methodological issue. Since there are so many uncertainties in defining scenarios, probably this paper should have been done in probabilistic terms, especially considering that the UN is making population probabilistic projections that can be accessed.

Reviewer #4:

Remarks to the Author:

This study presents estimates of the long-term impact of the COVID-19 pandemic on US population size and distribution by sex and age through 2060, accounting for the combined impact on mortality, fertility and migration. The analysis compares projected US population by age and sex from the 2022 UN World Population Prospects (WPP) with a counterfactual scenario estimating their hypothetical values in the absence of the pandemic (and assuming identical post-pandemic mortality, fertility and migration rates).

Relevance and significance

Original contribution that is of wider interest. To my knowledge this is the first study that tries to empirically assess a long-term impact of the pandemic on population growth and distribution in the US. However, there are issues pertaining to the accuracy and reliability of the data (and the underlying assumptions) used to estimate the "direct" impact of the pandemic in 2020-22.

Main strengths

As the impact of the coronavirus pandemic on population trends went far beyond the upswings in mortality, a broader assessment of both short-term and the long-term "scars" the pandemic is likely to leave on population size and structure is valuable. This paper presents the main results in a succinct fashion, estimating the contribution of the disruptions in each component to the estimated population "losses" and presenting their age structure trends. It clearly shows that, in the long term, lower immigration due to pandemic-related restrictions is more consequential for population trends than the temporary upswing in mortality. It also shows that the impact of the pandemic on population age structure is minor. The results are well presented and the readers get a clear picture of the long-term "disruption" of the pandemic for absolute and relative population at different ages.

Main weaknesses

A proper assessment the pandemic's impact on long-term population trends critically depends on the quality of underlying data and estimates. It is key to properly capture the actual population trends and, more important, to provide plausible and well-justified hypothetical scenario of the likely population trends in the absence of the pandemic. The main weakness of this study relates to the data, estimates and assumptions pertaining to the pandemic and the early post-pandemic period, in 2020-2024. The data-related issues can be broken down into three distinct (though interrelated) concerns

- 1) Relying on the UN WPP data for assessing the actual population trends and estimating hypothetical population trends in 2020-2024;
- 2) Dealing with uncertainties about the actual population trends in the later stages of the pandemic in 2021-2022;
- 3) Addressing uncertainties in properly estimating the pandemic disruption to fertility, mortality and migration, both in terms of properly identifying the period when the pandemic affected these trends and in the actual "attribution" of the COVID-related impact on these trends

To specify these concerns in more detail:

Relying on UN WPP data

UN WPP provide detailed and standardised datasets on population trends, structures and components of population trends in the past (1950-2021) and in the future (2022-2100) for all countries worldwide. In this way, they offer reliable, highly standardised and easily accessible data. But this convenience also has drawbacks. First, the data may not provide sufficient level of detail needed to properly account for the impact of the COVID-19 disruption for population trends. This includes a yearly format of all the published data, which does not allow a finer temporal evaluation of the COVID-19 impact on population and forces the authors to work with annual trends only. Second, the UN data experts do not have sufficient capacity to provide the most up-to-date estimates for each country and for all the data published in UN WPP. This means that the latest WPP published in July 2022 partly relies on models and preliminary estimates especially for the 2021 data, when the COVID-19

pandemic was still in a full swing. Fertility rates by age, mortality rates by age and sex, and migration rates by age and sex, especially in 2021-22, are partly model-based rather than directly derived from the official vital statistics records.

Uncertainties about population trends in 2020-24

Partly related to the reasons outlined above, this study may not provide a precise picture of the actual trends in births, deaths and migration in 2021-2022, which could in turn bias estimates of the actual impact of the COVID-19 pandemic. For instance, the UN WPP estimated total number of deaths in the US in 2021 at 3.281 million, whereas the officially reported preliminary count published by the National Centre for Health Statistics amounts to 3.464 million. (NCHS Data Brief No. 456, December 2022). Getting the population and vital statistics trends in 2021-22 right is essential for the accuracy of the presented results, although it also means that the study needs to move beyond simply taking the UN WPP as a “benchmark” against which the alternative counterfactual scenario without covid impact could be evaluated.

Uncertainties about assessing the covid impact on fertility, mortality and migration

The most challenging analytical issue is to properly estimate the “direct” impact of covid-19 on population trends during the pandemic. This implies a need to properly identify the period when the pandemic has been impacting fertility, mortality, and migration. In addition, it also means the impact of covid should be distinguished from the influence of other factors and fluctuations on fertility, mortality and migration.

This study assumes that the direct COVID impact can be identified as a temporary disruption over specific years—lasting longest for mortality—2020-24, impacting migration in 2020-21, and fertility only in 2020. The underlying assumption that COVID-19 impacted population trends only temporarily (and thereafter fertility, mortality, and international migration would return to its pre-COVID trajectory thereafter) made it possible to derive internally consistent counterfactual scenarios of the population trends with and without the pandemic, using solely the UN WPP data. However, the temporal delineation of the COVID impact is arguably the most problematic aspect of this study. The 2020 fertility decline was linked to the covid-19 pandemic only in the last months of the year (pregnancies conceived in the early stage of the pandemic, in March 2020, resulted in live births in the last months of the year, mostly November-December). Thus, most of the birth decline in 2020 cannot be attributed to the pandemic and the pandemic impact can properly be estimated only when using monthly data and accounting for the pre-pandemic birth dynamics (births until October 2020). When pre-pandemic birth trends are accounted for, the number of births in 2021 then actually exceeded the expected number, resulting in a temporary pandemic “surplus” of births, and creating larger-than-expected size of the 2021 cohort (see paper by Bailey et al. 2022 cited in the study). Mortality rates were affected by covid-19 almost throughout the entire year 2020 and thereafter, so estimating the covid impact on deaths should be easier: taking “excess deaths” as a simple and suitable indicator looks warranted. Still, I would like to see a bit more detailed consideration of other factors, including drug overdose trends, that might have helped fuelling rising mortality trends since 2020, and that may make simple analyses based on excess deaths less suited for the assessment of the covid-19 impact on deaths. Finally, a lot of uncertainty pertains to the covid-19 impact on international migration data and trends. One reason is that international migration is highly volatile, even in relatively stable times. This study assumes that the UN-based projection of international migration since 2022 (around 1 million net migrants per year) is unaffected by the covid-19. At the same time, in 2022 international migration to the US was still impacted by a continuation of “Title 42” restrictive legislation, presumably rolled out in response to the pandemic—therefore, it is likely, that the long “arm” of covid-related legislation is negatively impacting immigration to the US to date—a factor not built in the presented scenarios.

Methods and reproducibility

The methodology (cohort component projection model to project alternative scenarios of population change) is well established and clearly explained.

Summary, recommendations

This is a relevant and useful study. However, the authors need to address the data-related imitations discussed above. They should either complement the UN-based data and estimates with additional, more detailed and more up-to-date datasets, or they should provide a more in-depth assessment on the limitations of UN WPPs estimates for the US for the pandemic years, 2020-2022 and discuss how well the UN dataset captures the actual trends in population by age and sex and vital statistics (births, deaths, international migration, fertility, mortality and migration rates). The authors also need to provide a more nuanced assessment of the impact of the pandemic on births, deaths and international migration in the US, both in terms of a more precise period delineation (ideally moving beyond annual data format) and in terms of the accuracy of evidence, distinguishing the impact of the pandemic and pandemic-related measures from other non-related trends and fluctuations in the period.

Response to Reviews

Subject: Revision (NCOMMS-23-08414) entitled "The Long-term Effects of the COVID-19 Pandemic on U.S. Population Structure"

Dear Reviewers,

Thank you for reviewing our manuscript entitled "The Long-term Effects of the COVID-19 Pandemic on U.S. Population Structure". We appreciate the thorough and helpful review of our manuscript and are more than happy to address the expressed concerns.

The suggestions of the reviewers were extensive, but fair and resulted in a major revision of the article and inclusion of multiple new analyses, with many included in the main text and an expanded appendix. We are confident that given our extensive and serious revisions, we were able to meet the concerns of the Editor and reviewers.

We provide our detailed response below. On the following pages we first address several overarching concerns regarding (1) data, and (2) analyses. Then, we document our point-by-point response to the reviewers' comments, cross-referencing similar feedback accordingly. We indicate reviewer/editor feedback in text beginning "C," our responses in text beginning "R", and the manuscript changes in response in "quotes". We have indicated changes in the revised manuscript with red font. We believe that our manuscript is strengthened because of this review and hope that the editor and the reviewers agree.

We also point all to our repository, where all analytic data, scripts, graphs, and gifs are available: tinyurl.com/OSFCovPop. We additionally highlight the subfolder "/review", where all materials used to generate our new graphs (those in the manuscript and in the appendix) can be found.

We are of course open to any additional suggestions you might have and look forward to receiving your feedback.

Sincerely,
Authors

OVERARCHING CONCERNS

Overarching Concern One: Data

A common theme from the reviews centers around our decision to use the United Nations World Population Prospects (UNWPP) data. We break these concerns into: (1) limitations of UNWPP, and (2) alternative data sources.

The UNWPP is an excellent data source for population projections. As Reviewer 4 notes:

“UN WPP provide detailed and standardised datasets on population trends, structures and components of population trends in the past (1950-2021) and in the future (2022-2100) for all countries worldwide. In this way, they offer reliable, highly standardised and easily accessible data” (C24).

These data are widely used and trusted, both in widely consulted sources such as Our World in Data as well as in well-published research. Recent demographic publications using this data include projections of global patterns in adults having to care for parents and children simultaneously (Alburez-Gutierrez et al. 2021) as well as calculations of excess kin loss associated with excess mortality during the COVID-19 pandemic (Snyder et al. 2022). In the medical field, the data has been used in work projecting the economic burden of Alzheimer’s disease until 2050 (Nandi et al. 2022) and in projecting global child mortality estimates forward to 2030 (Sharrow et al. 2022). While all data sources have drawbacks, we do continue to advocate for the use of the UNWPP as the best available data for the purpose of our study.

First, we address limitations of the UNWPP as outlined by Reviewers. Reviewer 3 summarizes the expressed concerns succinctly:

“UN Population division is doing a great job, but they work with more than 200 countries and of course, they do not have enough men power to deal with each country in the same detail as the Bureau of Census can deal with the US” (C18)

Manpower limitations are a valid concern. While the UNWPP takes on the enormous feat of gathering and standardizing data for all countries, they base their standardization efforts on what they deem to be the most trustworthy national data sources. For the United States, data are in fact based on official estimates from the Census Bureau and National Vital Statistics System, among others. We direct reviewers to this website, which we think provides a great comprehensive overview of which data sources are compiled:

<https://population.un.org/wpp/DataSources/840>. We also summarize these sources in Appendix 3 (discussed below, page 4).

Related, Reviewer 4 asked for confirmation that complete data are used for 2021. This is confirmed at the above weblink, which reads: “Total fertility rate and age pattern of fertility based on: (a) official estimates of age-specific fertility rates through 2021 [...]” and “Life expectancy at birth and age pattern of mortality based on: (a) official estimates through 2017; (b) registered deaths by age and sex available through 2021 [...]” We also think that it is important to note that there is a lag on the publication of the “official estimates” mentioned here. Official mortality data (often in the form of life tables) from the United States are released with a lag of several years because they must combine final mortality statistics with

Medicare data (for people aged 66–99). Due to bureaucratic technicalities, the Medicare data takes several years to finalize, resulting in delays in the publication of official estimates. For example, the official 2017 estimates that the UNWPP use were published in June 2019 – and no new official data were released for 17 months, in November 2020.

To check for the robustness and quality of the data we use, we compare the mortality and fertility estimates used by the UNWPP, as input for their projections, with the official estimates (note: mortality estimates for 2021 are not yet final) from the US Vital Statistics offices (NVSS) and the Human Mortality/Fertility Databases. The differences between data sources are minor, with small underestimations in mortality and fertility by the UNWPP. These underestimations likely suggest that our projections underestimate the amount of missing people due to the pandemic. We include these comparison graphs in Appendix 1.

Reviewer 3 also writes “**These data [the UNWPP data] do not necessarily reflect the real situation better than the data from the US Bureau of the Census**” (C24; *italics indicate author added*). We do not dispute this. Instead, we reiterate that the UNWPP data come from the US Census Bureau. Importantly, and we elaborate on this further below (see R18), the US Census Bureau in their population and migration estimates (those linked by Reviewer 3) rely on *fiscal year* (July to June) and the UNWPP uses *calendar year* (January to December). Any observed differences between Census Bureau and UNWPP estimates of migration are likely attributable to this.

To assure us and the reviewers further on this point, we tested a modification of our projection with different census-based migration estimates in the baseline projections. We do this by transforming the fiscal year estimates from the Census Bureau to the calendar year form our other data is in and that our projection requires. Since the Census Bureau fiscal year (FY) tends to run from July 1 in year Y to June 30 in year Y+1, it divides the year exactly in the middle. We thus opt for the simplest and most assumption-free calculation of $IC_{CY}(Y)$ Immigration Count (calendar year Y):

$$IC_{CY}(Y) = \frac{[IC_{FY}(Y - 1, Y) + IC_{FY}(Y, Y + 1)]}{2}$$

We simply calculate the calendar year equivalent of the fiscal year numbers of the Census Bureau by adding up the two fiscal years a calendar year is split into and dividing them by two. Using these self-derived Census-equivalent data leads to nearly identical results to our original estimates. Given the closeness of the immigration count estimates, more complex interpolation strategies for transforming fiscal to calendar year estimates would necessarily yield similar results. A more thorough explanation of this can be found in our response R18, and in the newly created Appendix 2.

Second, we explored the possibility of using alternative datasets:

We explored the possibility of using the US Census Bureau projections, as they have a project similar to the UNWPP, called the “International Database” (IDB; see: <https://www.census.gov/programs-surveys/international-programs/about/idb.html>). However, the IDB projections were last updated in December 2020 for the United States (see: <https://www.census.gov/programs-surveys/international-programs/about/idb/countries-and-areas.html>). The IDB also makes some data assumptions that we are weary of, including

setting TFR to 1.84 for all years into the future. The last time TFR was 1.84 in the United States was in 2015, and it has been declining at a small rate since.

We also explored the possibility of using the Congressional Budget Office’s projections (CBO; see: <https://www.cbo.gov/system/files/2023-01/58612-Demographic-Outlook.pdf>). Our primary concern with these data is that they come without methodological appendix. CBO does not provide any information about sources of their data (though the downloadable Excel files regularly indicate that the data come from the “Congressional Budget Office”), nor does it provide any information about how projections are done (i.e., cohort component method as in both Census IDB and UNWPP). The lack of transparency around data and methods as well as the lack of use by the research community led us to decide to not include these data in our analyses.

Thus, the two main alternative population projection sources both come with significant shortcomings: the IDB lack recent projections and include unintuitive numbers and the CBO lack methodological and data transparency. For these reasons we prefer to continue using the UNWPP.

We do think we could have explained the data better in the manuscript, so we have expanded our “Methods and Materials” section to now read:

“The UNWPP data for the U.S. are compiled from several federal statistical agencies -- the US Census Bureau and the Vital Registration offices – in conjunction with international estimates. More detailed information on the UNWPP data is available in our Appendix and via the UNWPP methodology site, accessible via: <https://tinyurl.com/UNWPPMethodology>.” (Manuscript Page 8, Paragraph 1)

We also created Appendix 3, which contains an overview of the projection sources we explored: the UNWPP, the IDB, and the CBO. We hope that this comparison of methods, data sources, years, and projected values facilitates understanding for why we opted for using UNWPP data.

In sum, we hope that the reviewers are convinced that the UNWPP is the best possible source for population projections. To the best of our knowledge, there are no additional projection datasets suitable for our analyses. If the reviewers have further recommendations in this direction, we will do our best to replicate our analyses with them.

Overarching Concern Two: Analyses

The reviewers then raised several concerns related to our analytic approach. The primary concern seemed to center around our lack of uncertainty estimates, with Reviewer 2 explicitly asking for them: **“Does this method provide uncertainty (standard error) estimates? If not, can you somehow provide some standard errors for the results?”** (C16), and Reviewer 3 stating **“probably this paper should have been done in probabilistic terms”** (C20). We appreciate the reviewers raising this concern, as it has pushed us to amend our approach, code, and analyses accordingly. We think the paper is much strengthened for this.

Previously we used a deterministic approach: applying survival probabilities and fertility rates directly to the population counts and the derived person-years. For example, this meant multiplying the survival probability at a given age by the number of people at that age. We now take a stochastic approach for our projections. For mortality, we treat each projection step as a binomial experiment, where we set the number of trials n equal to the age-specific population count and the success probability p equal to the survival probability in the respective age group. For fertility, we treat the number of live births in each year as a random draw from a Poisson distribution, where we set the mean equal to the number of live births that would have resulted if the age-specific fertility rates had been applied deterministically to the population. We repeat our population projections 1,000 times and present the median value for each indicator in the manuscript. A stochastic rather than a deterministic approach allows us to generate confidence intervals. All our figures have been updated to reflect this change and now include dashed lines to indicate 95% confidence intervals. Reviewers will also likely notice that the axes on our updated graphs have been slightly modified. This is because in some year/scenario combinations, it is now possible for a confidence interval to cross zero. Additionally, all figure captions have been amended to make note of the confidence intervals.

We also updated our “Materials and Methods” section to reflect this change and included the following sentences:

“Throughout all analyses, we use a stochastic modeling approach. For mortality, we treat each projection step as a binomial experiment, where we set the number of trials n equal to the age-specific population count and the success probability p equal to the survival probability in the respective age group. For fertility, we treat the number of live births in each year as a random draw from a Poisson distribution, where we set the mean equal to the number of live births that would have resulted if the age-specific fertility rates had been applied deterministically to the population. We repeat our population projections 1,000 times and present the median value for each indicator, and in all figures the dashed lines represent the 95% confidence intervals.”
(Manuscript Page 8, Paragraph 3)

A second concern shared by several reviewers is a desire for alternative scenarios. For example, Reviewer 2 writes **“I am wondering if the paper will merit from some sensitivity analyses assessing different scenarios for the population projection, for instance now you mention that the findings are based on a medium scenario”** (C16). The reviewer is correct here that our analysis is based on the medium scenario estimated by the UNWPP.

Mortality, fertility, or migration scenarios other than the medium scenario would produce meaningful results in our application if we assumed those to apply in the baseline (with COVID-19) but not the counterfactual (without COVID-19) projections (or vice versa). However, following the UNWPP, we assume in our counterfactual projections that the U.S. will return to its without-COVID trajectory shortly after the pandemic. Thus, most fluctuations in population counts resulting from different mortality, fertility, and migration scenarios for the time after 2024 would cancel out in our calculations.

There are, however, two ways in which a different choice of scenarios might affect our results: First, a higher or lower mortality rate post-2024 would speed up or slow down the time scale by which the pandemic effects on the population structure ‘fade out’. Similarly, a higher or lower fertility rate post-2024 would affect the number of never-born children due to reproductive age mortality or missing migration into reproductive ages during the pandemic. Second, given our interpolation approach for generating mortality, fertility, and migration conditions in the counterfactual projections, assuming different post-COVID conditions would also affect the values for the counterfactual projections during the period 2020–2024. Thus, we agree with the reviewers that replicating our analyses on the alternative scenarios from the UNWPP, such as the lower and upper confidence bounds of their mortality and fertility estimates, would be helpful. Unfortunately, however, the UNWPP only publishes summary indicators (life expectancy at birth, total fertility rate) for these confidence bounds and not the underlying age-specific rates that are necessary for our approach (i.e., mortality and fertility rates).

Reviewer 2 (C18) raises some concerns with the UNWPP estimates of migration, which we note above in (Overarching Concern 1: Data). We re-estimated our analyses with these different migration estimates for the years 2020 and 2021.

REVIEWER COMMENTS

Reviewer #1 (Remarks to the Author):

C1. I read with great interest your article on "The Long-term Effects of the COVID-19 Pandemic on U.S. Population Structure". As the authors state, while there is ample evidence that the pandemic has disrupted mortality, fertility, and migration patterns independently, little is known about how the disruption of these processes jointly affected population counts and, consequently, future population structure.

I have a number of comments and suggestions that I hope will help to improve the quality of the paper. First I'll briefly mention my main concern before highlighting minor points:

R1. We thank the reviewer for their enthusiasm and detailed comments.

C2. Main concern:

How the population structure will be different in 2060 as a result of the COVID-19 pandemic is based on a lot of conjecture. The authors themselves assume that "following UNWPP ... mortality, fertility, and migration [will] return to their pre-pandemic trajectories after a few years". It is therefore unclear why the authors suppose in the same sentence that "the pandemic may continue to negatively affect the U.S. mortality, fertility, and migration environments well into the future" (lines 222-224). The first part of this statement is incongruent with the latter part. Indeed, evidence from elsewhere corroborates the temporality of the demographic consequences of the pandemic (see some examples below).

I would rather concentrate on shorter term effects of the pandemic, say until 2030 or 2040 at the most, using different possible scenarios, i.e. similar to the Spanish study the authors cited as well as another study that should have been mentioned, which produced post-covid scenarios for Australia based on possible short, moderate and long-term effects.

R2. There are a couple of excellent suggestions wrapped into these paragraphs. First, thanks for drawing our attention to our inconsistent language and assertions (original lines 222-224). Determining what will happen to mortality, fertility, and migration in the coming years is very much up for debate – related is the question of when the pandemic will end. We know from literature on the 1918 influenza that determining the end of a pandemic is difficult and often contested (Charters and Heitman 2021; Saglanmak et al. 2011). We adjust our language in the section the reviewer highlights. That section now reads:

“Third, following UNWPP, we assume that mortality, fertility, and migration return to their pre-pandemic trajectories after a few years. **There is inconclusive evidence about what signals the end of a pandemic or epidemic (34), so it is possible that the assumptions from UNWPP are incorrect. Should that be the case, and mortality continue to remain higher than expected, fertility, and/or migration continue to remain**

lower than expected, then our estimates represent an underestimation.” (Manuscript Page 7, Paragraph 1)

We appreciate the suggestion to narrow the time window of our analyses. Analytically, we agree with this suggestion. The further in time a projection goes, the more likely it is that there is statistical (and substantial) uncertainty, which makes narrower time windows informative. However, we currently prefer to keep our analyses through 2060 based on the following rationale. First, the central tenant of our argument is that in the long-term migration will have a larger effect on the population size and structure than mortality. Second, our focus on long-term trends offers a novel contribution to the existing literature. As the reviewer notes, existing studies explore short-term effects, and while we do the same, we also push scientific attention further to consider and to study long-term consequences of COVID-19 as well. Third, our expansion to 2060 is statistically meaningful: even with confidence intervals, we still are able to learn about long-term effects. In Appendix 4 we provide updated graphs that end at 2040, so that the reviewer may see them. Ultimately, we defer to the editor here about the time cut-off they prefer and will be happy to amend the manuscript accordingly.

Thanks also for drawing our attention to the Australia study (Wilson et al. 2021). We missed this in our initial write-up, and our revised manuscript now includes this. Please find this now incorporated in our second paragraph:

“[...] (see González Leonardo and Spijker 2022 (7) and Wilson et al. 2022 (8) for two exceptions, in Spain and Australia).” (Manuscript Page 2, Paragraph 2)

Then, we conducted a robustness check with alternative migration estimates. We also add confidence intervals. We describe these in detail in the beginning of our response, as this is a concern raised by several reviewers (see Page 4 above).

C3. What was missing:

In the abstract, the authors provide the main result, namely that "If the pandemic had not occurred, the population of the U.S. would have 2.1 million (0.63%) more people in 2025, and 1.7 million (0.44%) more people in 2060" yet there was no table or graph that showed a time trend (say 2010-2060) of the US population. In fact, similar to the Australian and Spanish studies it would have been helpful to have had graphs of each demographic component as well, comparing the UNWPP and counterfactual scenarios.

R3. We agree that summary figures of overall change in population size would be helpful. Accordingly, we have generated these for our main analyses, and for each of the analyses that isolate the contributions of mortality, fertility, and migration and have placed them in Appendix 5.

Minor comments:

C4. Lines 29-30 The authors mention that the TFR "declined by 4%, from 1.71 to 1.64, between 2019 and 2020 but "seemingly recovered in 2021"". Could you state the exact TFR for 2021 rather than using a value judgement ("seemingly recovered")?

R4. Thank you for correcting our language here, we have revised this sentence. Additionally, we have updated our citation for the 2021 rate to be the final estimates. Previously we had relied on estimates from the provisional NVSS report. The sentence now reads:

“The total fertility rate declined by 4% **in the first year of the pandemic (2019 to 2020)**, from 1.71 to 1.64, **and then increased to 1.66** in 2021.” (Manuscript Page 2, Paragraph 1)

C5. In lines 38-41 the authors state the "The rippling effects of the pandemic for the size and age-distribution of the U.S. population will continue to be observed because of the prolonged consequences of disruptions to all three demographic processes: mortality, fertility, and migration". How sure are the authors of the supposed prolonged consequences of each demographic component? In several countries, migration reached record levels in 2022, in part due to the war in Ukraine, while life expectancy has also recuperated. For instance, in the Netherlands, the total population grew more in 2022 than double than in 2019, i.e. the year prior to the start of the Covid-19 pandemic: <https://www.cbs.nl/en-gb/news/2023/01/population-growth-almost-doubled-in-2022>. Even if we disregard the influx of Ukrainians, the increase would have been greater than in 2019. Closer to the US, i.e. Canada, also saw record-high population growth in 2022 where migration was responsible for 95.9%: <https://www150.statcan.gc.ca/n1/daily-quotidien/230322/dq230322f-eng.htm> These are just 2 examples I checked, but I urge the authors to rephrase this based on what has already been observed in the US and elsewhere.

R5. This is an important comment, and we thank the reviewer for motivating us to be more nuanced in our discussions of the complexities and interrelatedness of demographic processes. We understand how our original wording in the manuscript could lead to more general misinterpretations about our analyses and our intent. Since we measure the consequences of pandemic-era disruptions and not prolonged consequences, we have re-written the following sentence:

“**If not reversed**, the rippling effects of the pandemic for the size and age-distribution of the U.S. population will continue to be observed because of **the pandemic-era** disruptions to all three demographic processes: mortality, fertility, and migration, and **the interdependent consequences therein**.” (Manuscript Page 2, Paragraph 2)

Throughout the manuscript, we also discuss and reflect on the peculiarities of the United States context. For example, we draw the reviewer’s attention to our discussion of Title 42 (in the main manuscript, 5th paragraph of the introduction). Title 42 halted much of the immigration into the United States during the pandemic. It was only recently repealed (11 May 2023). We also note that the Census Bureau estimates “50% reduction in net migration between July 2020 and June 2021, an estimate that is nearly uniformly distributed across migrant types – immigrant visas, work visas, student visas, and refugees and asylum seekers – followed by a return to pre-pandemic levels, **and possibly higher**, in 2022 (original manuscript, page 2).” Note that most recent estimates from the US Census Bureau anticipate rates that are even higher in 2022 than initial publications, and we have updated our manuscript to reflect this (<https://tinyurl.com/censumigration>).

We see secular events (e.g., war in Ukraine) influencing immigration in many nations during the pandemic years, including the United States (see again <https://tinyurl.com/censusmigration>). These events, while important for international migration estimates, would only be of concern for our modelling approach if they would not have happened without the pandemic. That is, the other increases in migration discussed by the reviewer, such as those due to the war in Ukraine, are not assumed to be triggered by the COVID pandemic. Thus, they would be included in both the baseline and the counterfactual projections and, hence, cancel each other out. Similarly, the increases in migration discussed for Canada seem to be related to a policy attempt to alleviate shortages in labor supply that are not or not only caused by the pandemic.

Pre-pandemic migration patterns in the United States suggest that net international migration increased from about 828 thousand people in 2010 to just above 1.2 million people in 2016. From 2016 though, net international migration began declining (2018 net rates were similar rate to 2010) (see Schacter et al., citation at end). This decline, Schacter et al. note, is driven by stalling immigration of foreign-born individuals and increasing emigration of foreign-born individuals.

C6. 88-89 Do the authors really think that "increases in mortality dominate the pandemic's long-term effects on the dependency ratio ... will likely be visible until the late 2040s"? Please provide references from medical/public health literature to substantiate this claim.

R6. We have extensively checked, but thus far, no other work has projected the long-term consequences of the mortality disruption and calculated its effects on the dependency ratio. We believe that the ability to make such statements is a unique contribution of our projection method.

To make it clearer that this statement is the result of our calculations, rather than a belief derived from the literature, we reformulated our text from “We emphasize three important points” to “We emphasize three important **findings from our projection**”. (Manuscript Page 3, Paragraph 3)

To provide additional intuition for the claim, we direct the reviewer to Panel B of Figure 4, which displays the change in the old-age dependency ratio due to mortality changes and shows that it remains lower (-0.18) in 2040 due to mortality effects. The pyramid displaying missing population due to mortality in the first panel of Figure 3 demonstrates that the missing population among the elderly in 2040 is the driver of these changes in the dependency ratio. The simple intuition here is that changes in the elderly population have an outsized effect on the old-age dependency ratio, and that many of the people who died in the pandemic were between ages of 45-70. This is what the model picks up on, projects forward, and consequently produces our findings on the dependency ratio.

We also direct the reviewer to our response below (R7) about the old-age dependency ratio. They will see that we have now better acknowledged the limitations of this measure and that we have added Figure 5 to the main manuscript, where we show the magnitude of the change in the population share in each age group.

C7. 143-144 I found the explanation that "A smaller dependency ratio means that there are fewer dependents per working-aged person, so this result can be interpreted as meaning that the pandemic contributed to fewer people in need of financial and social support" rather simplistic, basically because utilizing the old-age dependency ratio (OADR) as an exclusive gauge for financial burden in the aftermath of the pandemic undermines the consideration of other significant variables pertinent to economic productivity. While the OADR has been commonly used to measure the proportion of the elderly population to the working-age population, it fails to account for other essential factors such as changes in labor force participation, unemployment, and associated benefits that affects the denominator here. Neglecting these critical variables can lead to inaccurate estimations of the consequences of the pandemic on the economy. Other, more useful, indicators, include Pension Worker Ratio (1), the ratio of non-workers to workers irrespective of their age (2), or, even better, the 'Non-Working-aged' Dependency Ratio, which is the number of non-working persons over 60 per full-time equivalent worker (3).

R7. We thank the reviewer for their suggestion of different measures of dependency ratios. We also appreciated their inclusion of several references to guide us there; this was helpful. Upon looking through these alternatives to OADR, though, we note that it is necessary to have the number/proportion of working and non-working people. Unfortunately for us, UNWPP does not project this, limiting us to the simple OADR. In recognition that this is not ideal, we have discussed this in our limitations:

“Additionally, we note that our calculations of dependency ratio are relatively simplistic. More nuanced calculations of dependency ratios (e.g., the “non-Working-aged” dependency ratio) necessitate estimations of the number of working vs. non-working people at each age (33–35), and this data are not available in projected form from the UNWPP.” (Manuscript Page 6, Paragraph 2)

, where citations 33-35 are now the references provided by Reviewer 1.

We recognize that the reviewer may not be fond of us including OADR at all, so we have made an alternative figure that shows the projected difference (baseline – counterfactual) in population share at each age group, and across all scenarios. This figure is now Figure 5 in our main manuscript. We hope that the reviewer finds this new figure useful for contextualizing the nature and magnitude of the change.

C8. 174-175 While the author's result "counterintuitively highlights that the pandemic's effect on migration is more consequential for population size than its effect on mortality", it is consistent with the Spanish study that the authors cited earlier on (ref 7).

R8. We have edited this sentence to note that our finding is consistent with the Spanish study, and thus also removed the word “counterintuitively”. The sentence now reads:

“This result highlights that the pandemic's effect on migration is more consequential for population size than its effect on mortality, a finding that is consistent with a similar study on Spain.” (Manuscript Page 5, Paragraph 4)

C9. 178-180 The authors state that "in the next four decades there will likely be fewer reproductive-aged (15-49 years old) people in the U.S. This is a result of pandemic deaths and fewer migrants in childbearing age, as well as of the second-order implications this has for never-born children". Sure, it this was partly due to pandemic deaths in childbearing age but the % of mortality from 15-49 year olds was very very low. I don't know how low in the US as I couldn't find it in the OSF repository link but is probably less than 5% (based on own published research I can say it was 3% in Spain in 2020)). So, second-order implications would be really minimal.

R9. Thanks for drawing our attention to this. We do not want to exaggerate our findings, so we have softened our language. The sentence now reads:

“This is a result of fewer migrants in childbearing ages, as well as, to a lesser extent, pandemic deaths and second-order implications of migration and mortality for never-born children.” (Manuscript Page 5, Paragraph 5)

C10. 229-231. The authors state that "The elevated pandemic mortality among population groups with traditionally high fertility, such as non-Hispanic Black and Hispanic people, and the reduced in-migration of foreign-born individuals, who usually have higher fertility than native-born individuals, imply that fertility rates may remain depressed for an extended time period". This statement appears to be more on conjecture than facts. I'm not sure what the authors consider as being "high fertility" but both African American and Hispanic fertility rates have dropped quite considerably in recent decades, and are only marginally above those of White non-Hispanics (for a comparison between 2010 and 2020 see e.g. <https://www.newgeography.com/content/007528-us-total-fertility-rates-toward-europe>). Please also provide a reference for the claim that "foreign-born individuals usually have higher fertility than native-born individuals".

R10. The reviewer is correct that our language is a bit problematic here. While non-Hispanic Black and Hispanic birthing people do have general fertility rates that are higher than the non-Hispanic white population (Martin et al. 2021), we are cognizant that as the sentence was previously written, it inadvertently passed a value judgement. In the United States, this is particularly problematic because of the history of forced and coerced sterilization of Black birthing people, done in effort to reduce their number of children (see Roberts 1999 for a historical and legal history of this). We do not wish to contribute to this problematic narrative and have thus removed this sentence entirely.

Reviewer #2 (Remarks to the Author):

C11. The above-mentioned manuscript examines the effect of the COVID-19 pandemic on the US population structure. The authors have used a cohort component projection method and have compared the population projections with and without the pandemic. In general, I enjoyed reading the paper and found the results novel. I have the following comments:

R11. We appreciate the supportive comments and strive to answer them in kind below.

C12. I find it hard to understand the plots and maybe part of it is that I do not understand the cohort component projection method. In particular:

- a) **Is the shaded area in Figure 1 uncertainty? How is this calculated? Why is there an upper bound on 0?**
- b) **Is there a way to incorporate uncertainty in Figures 2-4?**
- c) **I do not understand how the dependency ratio can be mortality, fertility and migration specific. Can you provide more information on the caption?**

R12. Thank you for pushing us to be clearer in our figures. We address each of these comments in order here:

- (a) We apologize for the confusion created by this figure. The shaded area is the absolute (panel a) and relative (panel b) difference between the baseline and counterfactual estimates. Without COVID (counterfactual) there would be more people at all ages, so there is always a loss of population. In the new version of this graph, we have included dashed lines for the 95% confidence intervals. We also updated the caption to this figure, and for all others, to note this: **“Dashed lines represent 95% confidence intervals.”**
- (b) Great question, and one that has been raised by the other reviewers. All figures now include a dashed line to show the 95% confidence intervals. See above (Overarching Concern 2: Analyses) for our explanation of these.
- (c) Sorry about the confusion here. We have added the following sentence to the caption for Figure 4, in hopes that it clarifies this:

“Estimates for “mortality” indicate the projected difference in baseline vs. counterfactual dependency ratio if only mortality had changed during the pandemic, “fertility” if only fertility had changed, and “migration” if only migration had changed.”

C13. I think the methodology merits further elaboration

- a) **I think you need to provide more information about the dependency ratio.**
- b) **I think there is little information about the cohort component projection method and the paper cannot stand alone without reading about the method independently. I would suggest to provide the maths and more information about this method. Does this method provide uncertainty (standard error)**

estimates? If not, can you somehow provide some standard errors for the results?

- c) I am wondering if the paper will merit from some sensitivity analyses assessing different scenarios for the population projection, for instance now you mention that the findings are based on a medium scenario. Can you also check the variation of the results with respect to different migration schedules?

R13. There are several great suggestions here and we address each below:

- (a) We draw the reviewer's attention to our main manuscript, in "Materials and Methods," where we describe the dependency ratio calculations:

"We calculate total dependency ratios for the baseline and counterfactual scenarios as the ratio between non-working aged persons (younger than age 15 and older than age 64) and working aged persons (ages 15-64). Old-age dependency ratio is the ratio of the number of persons aged 65 and older to persons aged 15-64; young-age dependency ratio is the ratio of the number of persons younger than age 15 to persons aged 15-64."

Moreover, in response to C7 above, we have included some additional limitations in the "Materials and Methods" section around our use of the old-age dependency ratio:

"Additionally, we note that our calculations of dependency ratio are relatively simplistic. More nuanced calculations of dependency ratios (e.g., the "non-Working-aged" dependency ratio) necessitate estimations of the number of working vs. non-working people at each age (33-35), and this data are not available in projected form from the UNWPP." (Manuscript Page 6, Paragraph 2)

We hope that these address the reviewer's concern around information on the dependency ratio, and if not, we would appreciate a bit of guidance around what components could use clarification.

- (b) We are sorry the reviewer found the cohort component method unclear. We recognize that not all readers might be familiar with the method, and that we ought to be as clear as possible. We would, however, like to draw the reviewer's attention to a comment from Reviewer 4: "**The methodology (cohort component projection model to project alternative scenarios of population change) is well established and clearly explained**" (C24). To reconcile C13b with C24, we opt to include some additional information about the method in Appendix 6, including equations. As to the reviewer's suggestion for standard errors: thanks to encouragement here and from other reviewers, we now include 95% confidence intervals around our estimates. Please see above, in our overarching comment #2 (Analyses), for a thorough explanation of this.
- (c) Thanks, the reviewer is correct that we base our estimates on the medium scenario from the UNWPP. We are unable to replicate analyses on the other scenarios conducted by the UNWPP because the underlying rates in those models (i.e., age-

specific mortality rates, and age-specific fertility rates) are not available; only the summary indicators (life expectancy at birth, total fertility rate) are.

In the absence of age- and sex-specific migration counts for the U.S., we redistribute total net migration counts for the U.S. across age and sex using model migration schedules developed by Andrei Rogers and Luis Castro. Specifically, in line with UNWPP, we use the “family” migration schedule for our analysis. The two other available model schedules are “female labor” (used by UNWPP for Bahamas, Hong Kong, Macao, and Guyana) and “male labor” (used by UNWPP for Antigua and Barbuda, Azerbaijan, Bangladesh, Barbados, Belize, Botswana, British Virgin Islands, Burkina Faso, Cambodia, Curacao, Egypt, Equatorial Guinea, Gabon, Grenada, Guadeloupe, Guinea, Kuwait, Maldives, Malta, Mozambique, Nepal, Pakistan, Saudi Arabia, Senegal, Seychelles, Solomon Islands, South Africa, Suriname, Trinidad and Tobago, Tunisia, United Republic of Tanzania, Uzbekistan, Wallis and Futuna Islands, Western Sahara, Zambia). As the reviewer will see from the listed countries, assuming a “female” or “male labor” migration schedule seems implausible for a country of the Global North such as the U.S. We are thus hesitant to run additional models with alternative model migration schedules. However, in response to Reviewer 3 (C15), we re-estimated our baseline model with slightly different net migration counts. These are available below (R15) and in Appendix 2. We elaborate further on different scenarios in our overarching comment #2 (Analyses).

Reviewer #3 (Remarks to the Author):

C14. The paper is a projection exercise based on the UN WPP2022 revision. UN Population Division in its forecasts tried to incorporate available information on the consequences of COVID-19 in defining scenarios for future population trends. The authors tried to modify UN scenarios for mortality, fertility, and migration in such a way that from their point of view would eliminate the consequences of COVID-19 to measure the net effect of COVID-19 on the US population structure.

In any population projection, the resultant population composition is fully defined by scenarios of 3 components of population change – fertility, mortality, and migration. It makes sense to discuss the results if projection scenarios make sense. Unfortunately, I am not convinced of the meaningfulness of counterfactual scenarios.

I see this paper as just a pure exercise that is irrelevant to a real-world situation. I do not see any justification for counterfactual scenarios.

R14. We are sorry to hear that the reviewer does not see the merit in counterfactual scenarios and, as such, is seemingly unconvinced of the meaningfulness of our manuscript. While we are unsure that any explanation from us may change the reviewers mind about this, we still attempt to do so here.

First, we think it is important to highlight the utility of counterfactual scenarios as analytic tools. These are incredibly common and immensely useful in the social sciences (for some recent summary pieces, see: Mahoney and Barrenechea 2019; Morgan and Winship 2015; Rohlfing and Zuber 2021). We agree with the reviewer that such analyses require advanced statistical scrutiny, so we further strengthened our manuscript by conducting additional analyses and robustness checks (replication where applicable and possible). We also now provide our estimates with confidence intervals to account for stochastic uncertainty.

Second, as evidence of the importance of counterfactual scenarios, we draw the reviewer's attention to a few recent studies which have relied on similar counterfactual analyses to isolate the effects of the COVID-19 pandemic on population size and structure (Gonzalez-Leonardo and Spijker 2022; Wilson et al. 2022), fertility rates and number of births (Bailey et al. 2022), preterm birth rates (Margerison et al. 2022), and a more general study that applies counterfactual scenarios to understand how many (U.S.) Americans would be alive if the U.S. had the mortality rate of other wealthy nations (Bor et al. 2023). Our major contribution, as we see it, is studying all three of the components of population change (mortality, fertility, and migration) to provide a larger and more nuanced picture of the consequences of the pandemic. Ultimately we see counterfactual thinking as one of the key tools in trying to isolate pathways linking the past to the present ("If we had not given those persons medication X, then they would have higher probability of Y symptoms") and the present to the future, insofar as we engage in projections of it ("If we assume Chinese TFR will recover to 2.0, then we would project a population of Z size in 2050"). We therefore see it as well-suited to understand how the pandemic changed current and future population structure.

As to the reviewer's comment that our work is "**just a pure exercise that is irrelevant to a real-world situation,**" we contend that our work is very much rooted in a real-world situation: a pandemic, that changed mortality rates, fertility behavior, and immigration patterns, all of which had direct effects on people's lives. Understanding the channels through

which a pandemic affects a country's population structure, now and into the future, is an important scientific endeavor. Anything helping us improve forecasting tools is useful, and incorporating and understanding how forecasts change with respect to events is a useful part of that process.

We do not want to overstate our contributions in any way, but we did not know how to better respond to a generic claim of uselessness than to reiterate the practical reality and consequences of the pandemic, the need to understand how it continues to affect population structure, and the value of having good forecasting tools for population.

The reviewer raises several specific methodological points, which we found quite useful, and we address these in kind below.

C15. First of all, why are UN data taken as the baseline? These data do not necessarily reflect the real situation better than the data from the US Bureau of the Census. UN Population division is doing a great job, but they work with more than 200 countries and of course, they do not have enough men power to deal with each country in the same detail as the Bureau of Census can deal with the US. For example, the UN estimated net migration to the US in 2021 as about 560 thousand migrants. U.S. Census Bureau estimated Net international migration between 2021 and 2022 above 1 million. <https://www.census.gov/newsroom/press-releases/2022/2022-population-estimates.html> Only this fact already undermines the conclusions of the paper.

R15. We thank the reviewer for their attention to detail regarding data sources. However, it is worth highlighting that the UNWPP uses Census Bureau estimates in their input data. We discuss this at length in our general comments above (Page 3), but also expand on this a bit more here.

Regarding the reviewer's point about inconsistent estimates between the two data sources for migration estimates, there is one crucial discrepancy: these two sources estimate net migration with different months. The UN relies on calendar year, Jan-Dec estimates (e.g., their estimates for 2021 are for January 2021 until December 2021), while the US Census Bureau uses fiscal year, July to June estimates (e.g., the 2021/2022 estimates for the Census Bureau are for July 2021 until June 2022).

We also thank the reviewer for pointing us towards the most recent estimates of U.S. net international migration provided by the Census Bureau (<https://tinyurl.com/censusmigration> and in the link provided by the reviewer), and we apologize for not referencing them in our original manuscript. We noticed that, compared to the numbers cited in our original manuscript (page 2), the Census Bureau adjusted its net migration estimates for 2010–2021 upward by a substantial amount. This revision alone, caused first and foremost by methodological changes, highlights the contested nature of defining and measuring international migration in the absence of administrative data, an issue that is widely acknowledged in the scientific literature on this topic (for examples, Abdel and Cohen 2019; Billsborrow 1997).

After revising its estimates, the Census Bureau now estimates US net international migration as follows:

July 2019-June 2020: 726,000 people
July 2020-June 2021: 376,000 people
July 2021- June 2022: 1,011,000 people

Using the simple approximation in the equation from Page 3 of this document, we convert these fiscal year estimates into calendar year estimates, resulting in migration estimates that are:

January-December 2020: 551,000 people
January-December 2021: 693,500 people

To compare these, our net migration estimates from the UNWPP (our input data) are:

January-December 2020: 675,560 people
January-December 2021: 561,580 people

As further comparison, we re-estimate our analyses using our calendar-year estimates of Census Bureau migration values, for the years 2020 and 2021, in the baseline projections. We document our process for transforming fiscal-year into calendar-year estimates above (in “Overarching Concern One: Data”). For the counterfactual projections, we keep our estimates of 1,105,143 (2020) and 1,051,841 (2021) people, which are almost identical to the Census Bureau estimate for net international migration post-COVID. The results for this robustness check are shown in Appendix 2. We present two figures to compare our original results with those from the robustness check. Appendix Figure 5 is a replication of Figure 2 A from the main text, but with the migration values in the baseline projection replaced for 2020 and 2021. Appendix Figure 6 is an exact replication of Figure 2 A from the main text. These figures are nearly identical in shape.

We hope that following this explanation we were able to reassure the reviewer that the migration input used by the UNWPP does not show major deviations from the estimates from the Census Bureau.

We also highlight that we updated the citations and main manuscript to reference the most recent net international migration estimates provided by the Census Bureau.

“Net migration between July 2020 and June 2021 was estimated to be 376,000, nearly half of what it was between 2019 and 2020, and 70% lower than the highest estimate of the preceding decade – 1,236,000 between 2015 and 2016 (6, 7).” (Manuscript Page 2, Paragraph 1)

C16. Why do the authors expect that COVID-19 had a negative effect on fertility and adjust TFR for 2021 to inflate it? It could very well be that fertility without COVID in 2021 could have been even lower than in 2019 if the TFR declining trend that began around 2007-2008 continued.

Mortality scenarios cause questions as well. CDC estimates life expectancy for males and females in 2021 as 73.2 and 79.1 years of life.

(https://www.cdc.gov/nchs/pressroom/nchs_press_releases/2022/20220831.htm) The corresponding estimates of the UN are more than 1 year higher - 74.3 and 80.2 years. UN scenarios for life expectancy for males and females do not foresee any “harvesting” effect at least in the medium scenario used in the paper. However, it is well possible that

after periods with excess mortality, there could be a period of mortality decline in subsequent times because the most fragile individuals passed away. For example, a very large number of COVID-19 deaths occurred in nursery homes. In many cases, COVID-19 just sped up the deaths that would have occurred soon anyway since life expectancy in nursery homes is very short. On the other hand, as the authors also noted no one knows about the long-term effects of COVID-19 and its effects on the humans who experienced the disease. So this is more a probabilistic issue.

R16. Thanks to the reviewer for their careful read and attention to detail here. We first draw attention to our manuscript (Manuscript Page 6, Paragraph 3), where we write:

“Moreover, the published UNWPP forecasts for the year 2022 correspond well with preliminary estimates of mortality, fertility, and migration (4, 6, 18) generating further trust in our baseline and counterfactual estimates for the pandemic period.”

That is to say, the projections generated by UNWPP align with what is reported by the US Vital Statistics offices. Evidence from the US shows that there was a fertility decline at the end of 2020, an increase in 2021, and then stability in 2022 (Bailey et al. 2022; Kearney and Levine 2023; Sobotka et al. 2023). There is also not enough support for a positive effect of COVID-19 in fertility intentions studies: the trends remain the same, or even show that COVID-19 is associated with a decrease in the likelihood of planning to have children (Luppi et al. 2020).

We would also like to highlight that, for the year 2020 in the counterfactual projections, we used the average TFR of the 2019 and 2021 TFRs provided by UNWPP. We distributed this TFR across maternal ages according to the 2021 age pattern of fertility. UNWPP reports the following TFR values for 2019 to 2021, which are close to the final data reported by NVSS (Osterman et al. 2019):

2019: 1.6892
2020: 1.6428
2021: 1.6619

Averaging the 2019 and 2021 TFRs, we obtain a counterfactual TFR of 1.67555. To put this into context, a regression-based imputation of the 2020 TFR based on 2010–2019 trends would give a counterfactual TFR of 1.685473. Thus, our approach of averaging 2019 and 2021 TFRs to obtain the 2020 counterfactual TFR is actually more conservative than an approach based on pre-pandemic trends.

We also thank the reviewer for highlighting the differences in life expectancy estimates between NVSS and UNWPP. In Appendix 1, we provide a detailed comparison between age-specific death rates (m_x) and age-specific probabilities of death (q_x) from life tables published by NVSS, Human Mortality Database, and UNWPP.

Overall, two main patterns are noticeable: First, there is strong agreement between NVSS, HMD, and UNWPP among ages below 90. Above age 90, UNWPP estimates are lower than NVSS and HMD estimates. Second, UNWPP provides lower mortality estimates already in 2019. Regarding the first point, we note that, according to our estimates, the largest mortality effects of the COVID-19 pandemic are always found below age 90, i.e., among ages for which mortality estimates in NVSS, HMD, and UNWPP are virtually identical. Regarding

the second point, the differences in 2019 estimates suggest general methodological differences between UNWPP on the one side and NVSS and HMD on the other side that are not COVID-19 specific. In fact, as we mention elsewhere, UNWPP relies on these official sources to generate their mortality estimates for 2021.

Finally, we agree with the reviewer about the complexity of quantifying the long-term effects of COVID-19. We do emphasize that we are quantifying the direct consequences of the pandemic, and the indirect consequences, such as mortality displacement, are beyond the scope of the present analyses. Reviewer 1 raised a similar concern about language, and to address their concern and the one articulated here, we have adjusted our language in the limitations of our paper to now read as:

“Third, following UNWPP, we assume that mortality, fertility, and migration return to their pre-pandemic trajectories after a few years. **There is inconclusive evidence about what signals the end of a pandemic or epidemic (34), so it is possible that the assumptions from UNWPP are incorrect. Should that be the case, and mortality continue to remain higher than expected, fertility, and/or migration continue to remain lower than expected, then our estimates represent an underestimation.**” (Manuscript Page 7, Paragraph 1)

C17. And that brings me to another methodological issue. Since there are so many uncertainties in defining scenarios, probably this paper should have been done in probabilistic terms, especially considering that the UN is making population probabilistic projections that can be accessed.

R17. This is a great suggestion. However, we reiterate that UNWPP only publishes probabilistic estimates of summary indicators (life expectancy at birth, total fertility rate), not the underlying age-specific rates that would be necessary for our projections. We followed the reviewer’s suggestion and conducted all analyses stochastically, with uncertainty added around the counts generated from the medium scenario (we elaborate on this above in our overarching concern 2, on Analyses). All figures have been updated to include 95% confidence intervals. Figure captions have also been updated to note this.

Reviewer #4 (Remarks to the Author):

C18. This study presents estimates of the long-term impact of the COVID-19 pandemic on US population size and distribution by sex and age through 2060, accounting for the combined impact on mortality, fertility and migration. The analysis compares projected US population by age and sex from the 2022 UN World Population Prospects (WPP) with a counterfactual scenario estimating their hypothetical values in the absence of the pandemic (and assuming identical post-pandemic mortality, fertility and migration rates).

Relevance and significance

Original contribution that is of wider interest. To my knowledge this is the first study that tries to empirically assess a long-term impact of the pandemic on population growth and distribution in the US. However, there are issues pertaining to the accuracy and reliability of the data (and the underlying assumptions) used to estimate the "direct" impact of the pandemic in 2020-22.

R18. We appreciate the reviewer's comments and critiques and address them each below.

C19. Main strengths

As the impact of the coronavirus pandemic on population trends went far beyond the upswings in mortality, a broader assessment of both short-term and the long-term "scars" the pandemic is likely to leave on population size and structure is valuable. This paper presents the main results in a succinct fashion, estimating the contribution of the disruptions in each component to the estimated population "losses" and presenting their age structure trends. It clearly shows that, in the long term, lower immigration due to pandemic-related restrictions is more consequential for population trends than the temporary upswing in mortality. It also shows that the impact of the pandemic on population age structure is minor. The results are well presented and the readers get a clear picture of the long-term "disruption" of the pandemic for absolute and relative population at different ages.

R19. Thank you for your clear summary of our paper. We agree that these are the paper's main strengths.

C20. Main weaknesses

A proper assessment the pandemic's impact on long-term population trends critically depends on the quality of underlying data and estimates. It is key to properly capture the actual population trends and, more important, to provide plausible and well-justified hypothetical scenario of the likely population trends in the absence of the pandemic. The main weakness of this study relates to the data, estimates and assumptions pertaining to the pandemic and the early post-pandemic period, in 2020-2024. The data-related issues can be broken down into three distinct (though interrelated) concerns

- 1) Relying on the UN WPP data for assessing the actual population trends and estimating hypothetical population trends in 2020-2024;**
- 2) Dealing with uncertainties about the actual population trends in the later stages of**

the pandemic in 2021-2022;

3) Addressing uncertainties in properly estimating the pandemic disruption to fertility, mortality and migration, both in terms of properly identifying the period when the pandemic affected these trends and in the actual “attribution” of the COVID-related impact on these trends

R20. Thank you for raising these concerns so thoroughly and succinctly. We answer them point by point:

C21. Relying on UN WPP data.

UN WPP provide detailed and standardised datasets on population trends, structures and components of population trends in the past (1950-2021) and in the future (2022-2100) for all countries worldwide. In this way, they offer reliable, highly standardised and easily accessible data. But this convenience also has drawbacks. First, the data may not provide sufficient level of detail needed to properly account for the impact of the COVID-19 disruption for population trends. This includes a yearly format of all the published data, which does not allow a finer temporal evaluation of the COVID-19 impact on population and forces the authors to work with annual trends only. Second, the UN data experts do not have sufficient capacity to provide the most up-to-date estimates for each country and for all the data published in UN WPP. This means that the latest WPP published in July 2022 partly relies on models and preliminary estimates especially for the 2021 data, when the COVID-19 pandemic was still in a full swing. Fertility rates by age, mortality rates by age and sex, and migration rates by age and sex, especially in 2021-22, are partly model-based rather than directly derived from the official vital statistics records.

R21. The UNWPP is the gold standard for world population projections. However, the reviewer is correct to point out that this comes with many limitations, including the two articulated here: (1) use of yearly data (rather than weekly or monthly), and (2) use of preliminary data for 2021/22 (rather than complete data). We address these concerns in order.

(1) The COVID-19 pandemic had an uneven temporal progression marked by waves -- for example, the omicron wave arrived in the United States in late 2021. This contributed to weekly/monthly mortality fluctuations with important research analyzing these patterns (for some examples, see: Cronin and Evans 2021; Lundberg et al. 2023; Rossen et al. 2020; Stokes et al. 2021; Weinberger et al. 2021). We chose to focus our analyses on yearly aggregate data, rather than to focus on these short-term temporal patterns. The reason for refraining from examining short-term patterns is rooted in the holistic approach of our work, which attempts to account for mortality, immigration and fertility changes in a joint framework. While monthly mortality and fertility data are available from CDC Wonder – an online query tool provided and maintained by the United States Department of Health and Human Services – a similar resource is not available for migration data. Migration data is difficult to collect and access, and hence there is no dataset documenting short-term migration patterns in and out of the U.S. The best option are the fiscal year estimates provided by the U.S. Census Bureau and the calendar year estimates from the UNWPP (see our response R18 for a discussion of comparing fiscal year vs. calendar year estimates). We

see the merit in smaller level temporal analyses, but it is unfortunately not feasible within the broader aim of this project. To acknowledge this, we now add a sentence to our discussion:

" While we are limited by the lack of migration data at smaller temporal windows (e.g., month or week), future work with better data availability might consider analyzing this to gain a more nuanced understanding of how these processes vary across other temporal dimensions." (Manuscript Page 6, Paragraph 4)

(2) refers to UNWPP use of preliminary data for 2021. We direct the reviewer to the previous "Overarching Concerns" section, where we discuss this at length. We provide a brief summary: the UNWPP state in their data documentation (see here for US: <https://population.un.org/wpp/DataSources/840>) that the mortality and fertility data come from registered deaths and births through 2021. To verify the quality of our input data we compared the UNWPP estimates with final estimates from US Vital Statistics, and the Human Mortality/Fertility Databases (another widely used data source). We show these comparisons in Appendix 1. The rates are incredibly close, with few exceptions above age 90 for mortality. We hope that the reviewer is reassured that the data underlying the model is of high quality.

C22. Uncertainties about population trends in 2020-24.

Partly related to the reasons outlined above, this study may not provide a precise picture of the actual trends in births, deaths and migration in 2021-2022, which could in turn bias estimates of the actual impact of the COVID-19 pandemic. For instance, the UN WPP estimated total number of deaths in the US in 2021 at 3.281 million, whereas the officially reported preliminary count published by the National Centre for Health Statistics amounts to 3.464 million. (NCHS Data Brief No. 456, December 2022). Getting the population and vital statistics trends in 2021-22 right is essential for the accuracy of the presented results, although it also means that the study needs to move beyond simply taking the UN WPP as a "benchmark" against which the alternative counterfactual scenario without covid impact could be evaluated.

R22. We are grateful that the reviewer highlights such an important concern. The UNWPP provides rigorous projections, both in its input data and methods. Yet, error is inevitable. In recognition of this, we have updated our analyses and now provide a more nuanced picture, where we account for such discrepancies through the notion of statistical uncertainties. Hence, we are now taking a more stochastic approach, rather than the deterministic one we followed in our initial submission. By doing so, we are now able to generate confidence intervals. We provide a more thorough explanation of this above (Page 5, Overarching Concern Two: Analyses) and in the manuscript on page 8.

C23. Uncertainties about assessing the covid impact on fertility, mortality and migration.

The most challenging analytical issue is to properly estimate the "direct" impact of covid-19 on population trends during the pandemic. This implies a need to properly

identify the period when the pandemic has been impacting fertility, mortality, and migration. In addition, it also means the impact of covid should be distinguished from the influence of other factors and fluctuations on fertility, mortality and migration.

This study assumes that the direct COVID impact can be identified as a temporary disruption over specific years—lasting longest for mortality—2020-24, impacting migration in 2020-21, and fertility only in 2020. The underlying assumption that COVID-19 impacted population trends only temporarily (and thereafter fertility, mortality, and international migration would return to its pre-COVID trajectory thereafter) made it possible to derive internally consistent counterfactual scenarios of the population trends with and without the pandemic, using solely the UN WPP data. However, the temporal delineation of the COVID impact is arguably the most problematic aspect of this study. The 2020 fertility decline was linked to the covid-19 pandemic only in the last months of the year (pregnancies conceived in the early stage of the pandemic, in March 2020, resulted in live births in the last months of the year, mostly November-December). Thus, most of the birth decline in 2020 cannot be attributed to the pandemic and the pandemic impact can properly be estimated only when using monthly data and accounting for the pre-pandemic birth dynamics (births until October 2020). When pre-pandemic birth trends are accounted for, the number of births in 2021 then actually exceeded the expected number, resulting in a temporary pandemic “surplus” of births, and creating larger-than-expected size of the 2021 cohort (see paper by Bailey et al. 2022 cited in the study). Mortality rates were affected by covid-19 almost throughout the entire year 2020 and thereafter, so estimating the covid impact on deaths should be easier: taking “excess deaths” as a simple and suitable indicator looks warranted. Still, I would like to see a bit more detailed consideration of other factors, including drug overdose trends, that might helped fuelling rising mortality trends since 2020, and that may make simple analyses based on excess deaths less suited for the assessment of the covid-19 impact on deaths. Finally, a lot of uncertainty pertains to the covid-19 impact on international migration data and trends. One reason is that international migration is highly volatile, even in relatively stable times. This study assumes that the UN-based projection of international migration since 2022 (around 1 million net migrants per year) is unaffected by the covid-19. At the same time, in 2022 international migration to the US was still impacted by a continuation of “Title 42” restrictive legislation, presumably rolled out in response to the pandemic—therefore, it is likely, that the long “arm” of covid-related legislation is negatively impacting immigration to the US to date—a factor not built in the presented scenarios.

R23. It is clear the reviewer has read our manuscript and analyses closely, and we greatly appreciate that. We want to first address the concern around estimating the identification of the COVID period. Scientists often debate what constitutes the end of a pandemic (Charters and Heitman 2021). For example, evidence and scientific conversations from the 1918 flu are still inconclusive about when it ended (Ansart et al. 2009; Chandra et al. 2021; Saglanmak et al. 2011). Yet, the recent evidence that there was a flu resurgence in 1919 and 1920 (in some regions) does not invalidate the analyses that examine it through the end of 1918. Instead, the results that analyzed the flu through 1918 are interpreted with the new limitation and caveat that perhaps their effects are underestimations.

Now, we apply the same logic from research on the 1918 flu here. We make an assumption about when the pandemic’s reach on mortality, fertility, and migration will end. The assumption we make is the same one that the UNWPP uses (and that the reviewer correctly

documents here). This assumption may not be right. We will not know for several years. However, it is still important to conduct these analyses and to make these projections. When future research emerges about the pandemic, we hope that researchers will expand on the analyses we have conducted here. And we hope that future analyses showing different “ends” of the pandemic (for mortality, fertility, and migration) will interpret our analyses with the same caveats that research on the 1918 flu is afforded.

As we note above (to Reviewer 1, in R2), we expand our limitations to further highlight this limitation:

“Third, following UNWPP, we assume that mortality, fertility, and migration return to their pre-pandemic trajectories after a few years. **There is inconclusive evidence about what signals the end of a pandemic or epidemic (34), so it is possible that the assumptions from UNWPP are incorrect. Should that be the case, and mortality continue to remain higher than expected, fertility, and/or migration continue to remain lower than expected, then our estimates represent an underestimation.**” (Manuscript Page 7, Paragraph 1)

We perceive the reviewer's suggestion to unpack different causes of COVID-related mortality and its effect on population projection as an important extension for research. However, it goes beyond this study and we instead motivate further research with such a scientific agenda in our discussion:

“Additionally, if the midlife mortality crisis in the U.S. persists (48, 49), and if rising mortality rates from the opioid epidemic are not curtailed, then deaths among reproductive-aged people will continue to rise, resulting in fewer people at young adult and midlife ages. Applying the cohort component projection method to these crises will be valuable for understanding the magnitude of their consequences for the U.S. population.”

We also agree with the reviewer about the limitations of international migration. We again draw attention to the additional analyses done with migration estimates (discussed at length in Overarching Concern 1: Data). It is also worth highlighting that at the time of writing, this Title 42 has been repealed, so the direct consequences attributable to it have likely waned.

Finally, we agree with the reviewer that the assumption of temporality (i.e., consistent rates across the entirety of a year) is challenging. We draw their attention to our response on this in C21. We think it is an important area of pursuit, but we are limited in our ability to do this. Also as noted above, we add a mention of this in our limitations:

“**While we are limited by the lack of migration data at smaller temporal windows (e.g., month or week), future work with better data availability might consider analyzing this to gain a more nuanced understanding of how these processes vary across other temporal dimensions.**” (Manuscript Page 6, Paragraph 4)

C24. Methods and reproducibility

The methodology (cohort component projection model to project alternative scenarios of population change) is well established

R24. Thank you.

C25. Summary, recommendations.

This is a relevant and useful study. However, the authors need to address the data-related limitations discussed above. They should either complement the UN-based data and estimates with additional, more detailed and more up-to-date datasets, or they should provide a more in-depth assessment on the limitations of UN WPPs estimates for the US for the pandemic years, 2020-2022 and discuss how well the UN dataset captures the actual trends in population by age and sex and vital statistics (births, deaths, international migration, fertility, mortality and migration rates). The authors also need to provide a more nuanced assessment of the impact of the pandemic on births, deaths and international migration in the US, both in terms of a more precise period delineation (ideally moving beyond annual data format) and in terms of the accuracy of evidence, distinguishing the impact of the pandemic and pandemic-related measures from other non-related trends and fluctuations in the period.

R25. We are glad the reviewer finds the study relevant and useful. We addressed the limitations with regard to data as extensive as we could throughout this document and updated manuscript and supplementary files accordingly. We also provide a detailed response to data-related limitations in the first *Overarching Concerns* section of this document.

Importantly, we have added various additional analyses to complement our estimates. First, we compared the mortality and fertility estimates used by the UNWPP with the official estimates from the US Vital Statistics offices (NVSS) and the Human Mortality/Fertility Databases. We found rather minor differences (Appendix 1), and this exercise in general points us to the likelihood of small underestimation of mortality and fertility projections based on UNWPP data. Moreover, we tested a modification of our projection with different census-based migration estimates. We do this by transforming the fiscal year estimates from the Census Bureau to the calendar year form. A more detailed explanation of this can be found in our response R15, and in the newly created Appendix 2.

Second, we explored the possibility to use other data sources and expanded our “Methods and Materials” section along with the inclusion of a new Appendix 3. We provide a comprehensive overview of the existing datasets (Supplementary Table 2): the UNWPP, the IDB, and the CBO. We compared their methods, data, and projection approaches. After this exercise, we are still convinced that the UNWPP is the most suitable source for us; moreover, there are simply no additional projection datasets suitable for our analyses. We also reflect on this in the “Methods and Materials” section:

“The UNWPP projections for the U.S. are based on data from several federal statistical agencies -- the US Census Bureau and the Vital Registration offices – in conjunction with international estimates. More detailed information on the UNWPP data is available in our Appendix and via the UNWPP methodology site, accessible via: <https://tinyurl.com/UNWPPMethodology>.” (Manuscript Page 8, Paragraph 1)

Third, another important extension of our analytic strategy is the adoption of a stochastic approach which accounts for stochastic uncertainty in the modelling. We are now able to

present our analyses with confidence intervals. This is an important update to our analytic approach, and we are immensely grateful to the reviewers for pushing us in that direction. See updated figures in the manuscript. We updated our “Materials and Methods” section to reflect this change and included the following sentences:

“Throughout all analyses, we use a stochastic modeling approach. For mortality, we treat each projection step as a binomial experiment, where we set the number of trials n equal to the age-specific population count and the success probability p equal to the survival probability in the respective age group. For fertility, we treat the number of live births in each year as a random draw from a Poisson distribution, where we set the mean equal to the number of live births that would have resulted if the age-specific fertility rates had been applied deterministically to the population. We repeat our population projections 1,000 times and present the median value for each indicator, and in all figures the dashed lines represent the 95% confidence intervals.”
(Manuscript Page 8, Paragraph 3)

Fourth, while we agree that moving beyond yearly data can be an important nuance, we chose to focus on the yearly aggregations for several reasons. We refer the reviewer to our detailed response R15 and R21 on this matter, and particularly note the data availability issues as well as methodological concerns. Overall, we hope that the scope of additional work we performed and extended reflections of the limitations of our study further convince the reviewer of the robustness of our study.

- Abel, G. J., & Cohen, J. E. (2019). Bilateral international migration flow estimates for 200 countries. *Scientific data*, 6(1), 82
- Alburez-Gutierrez, D., Mason, C., & Zagheni, E. (2021). The “sandwich generation” revisited: Global demographic drivers of care time demands. *Population and Development Review*, 47(4), 997-1023.
- Ansart, S., Pelat, C., Boelle, P. Y., Carrat, F., Flahault, A., & Valleron, A. J. (2009). Mortality burden of the 1918–1919 influenza pandemic in Europe. *Influenza and other respiratory viruses*, 3(3), 99-106.
- M. Bailey, J. Currie, H. Schwandt, “The Covid-19 Baby Bump: The Unexpected Increase in U.S. Fertility Rates in Response to the Pandemic” (National Bureau of Economic Research, 2022).
- Bilsborrow, R. E. (1997). *International migration statistics: Guidelines for improving data collection systems*. International Labour Organization.
- Bor, J., Stokes, A. C., Raifman, J., Venkataramani, A., Bassett, M. T., Himmelstein, D., & Woolhandler, S. (2023). Missing Americans: Early Death in the United States, 1933-2021. *PNAS*, <https://doi.org/10.1093/pnasnexus/pgad173>.
- Chandra, S., Christensen, J., Chandra, M., & Paneth, N. (2021). Pandemic reemergence and four waves of excess mortality coinciding with the 1918 influenza pandemic in Michigan: insights for COVID-19. *American Journal of Public Health*, 111(3), 430-437.
- Charters, E., & Heitman, K. (2021). How epidemics end. *Centaurus*, 63(1), 210-224.
- Cronin, C. J., & Evans, W. N. (2021). Excess mortality from COVID and non-COVID causes in minority populations. *Proceedings of the National Academy of Sciences*, 118(39), e2101386118.
- M. González-Leonardo, J. Spijker, The impact of Covid-19 on demographic components in Spain, 2020–31: A scenario approach. *Population Studies*, 1–17 (2022).
- Kearney, M.S., Levine, P.B. The US COVID-19 baby bust and rebound. *J Popul Econ*(2023). <https://doi.org/10.1007/s00148-023-00965-x>
- Lundberg, D. J., Wrigley-Field, E., Cho, A., Raquib, R., Nsoesie, E. O., Paglino, E., ... & Stokes, A. C. (2023). COVID-19 Mortality by Race and Ethnicity in US Metropolitan and Nonmetropolitan Areas, March 2020 to February 2022. *JAMA Network Open*, 6(5), e2311098-e2311098.
- Luppi, F., Arpino, B., & Rosina, A. (2020). The impact of COVID-19 on fertility plans in Italy, Germany, France, Spain, and the United Kingdom. *Demographic Research*, 43, 1399-1412.

Mahoney, J., & Barrenechea, R. (2019). The logic of counterfactual analysis in case-study explanation. *The British journal of sociology*, 70(1), 306-338.

Margerison, C. E., Bruckner, T. A., MacCallum-Bridges, C., Catalano, R., Casey, J. A., & Gemmill, A. (2023). Exposure to the early COVID-19 pandemic and early, moderate and overall preterm births in the United States: a conception cohort approach. *Paediatric and Perinatal Epidemiology*, 37(2), 104-112.

Martin JA, Hamilton BE, Osterman MJK. Births in the United States, 2020. NCHS Data Brief, no 418. Hyattsville, MD: National Center for Health Statistics. 2021. DOI: <https://dx.doi.org/10.15620/cdc:109213>.

Morgan, S. L., & Winship, C. (2015). *Counterfactuals and causal inference*. Cambridge University Press.

Nandi, Arindam & Counts, Nathaniel & Chen, Simiao & Seligman, Benjamin & Tortorice, Daniel & Vigo, Daniel & Bloom, David. (2022). Global and regional projections of the economic burden of Alzheimer's disease and related dementias from 2019 to 2050: A value of statistical life approach. *EClinicalMedicine*. 51. 101580. 10.1016/j.eclinm.2022.101580.

Osterman, M. J., Hamilton, B. E., Martin, J. A., Driscoll, A. K., & Valenzuela, C. P. (2023). Births: final data for 2021. Accessed via: <https://www.cdc.gov/nchs/data/nvsr/nvsr72/nvsr72-01.pdf>.

Roberts, D. (2014). *Killing the black body: Race, reproduction, and the meaning of liberty*. Vintage.

Rohlfing, I., & Zuber, C. I. (2021). Check your truth conditions! Clarifying the relationship between theories of causation and social science methods for causal inference. *Sociological Methods & Research*, 50(4), 1623-1659.

Rossen, L. M., Branum, A. M., Ahmad, F. B., Sutton, P., & Anderson, R. N. (2020). Excess deaths associated with COVID-19, by age and race and ethnicity—United States, January 26–October 3, 2020. *Morbidity and Mortality Weekly Report*, 69(42), 1522.

Saglanmak, N., Andreasen, V., Simonsen, L., Mølbak, K., Miller, M. A., & Viboud, C. (2011). Gradual changes in the age distribution of excess deaths in the years following the 1918 influenza pandemic in Copenhagen: using epidemiological evidence to detect antigenic drift. *Vaccine*, 29, B42-B48.

Schachter, P. Borsella, A. Knapp, New Population Estimates Show COVID-19 Pandemic Significantly Disrupted Migration Across Borders. *United States Census Bureau* (2021) (December 6, 2022). Accessed via: <https://www.census.gov/library/stories/2021/12/net-international-migration-at-lowest-levels-in-decades.html>

Snyder, M., Alburez-Gutierrez, D., Williams, I., & Zagheni, E. (2022). Estimates from 31 countries show the significant impact of COVID-19 excess mortality on the incidence of family bereavement. *Proceedings of the National Academy of Sciences*, 119(26), e2202686119.

Sharrow D, Hug L, You D, Alkema L, Black R, Cousens S, Croft T, Gaigbe-Togbe V, Gerland P, Guillot M, Hill K, Masquelier B, Mathers C, Pedersen J, Strong KL, Suzuki E, Wakefield J, Walker N; UN Inter-agency Group for Child Mortality Estimation and its Technical Advisory Group. Global, regional, and national trends in under-5 mortality between 1990 and 2019 with scenario-based projections until 2030: a systematic analysis by the UN Inter-agency Group for Child Mortality Estimation. *Lancet Glob Health*. 2022 Feb;10(2):e195-e206. doi: 10.1016/S2214-109X(21)00515-5. PMID: 35063111; PMCID: PMC8789561.

Sobotka, T., Zeman, K., Jasilioniene, A., Winkler-Dworak, M., Brzozowska, Z., Alustiza-Galarza, A., ... & Jdanov, D. (2023). Pandemic Roller-Coaster? Birth Trends in Higher-Income Countries During the COVID-19 Pandemic. *Population and Development Review*.

Stokes, A. C., Lundberg, D. J., Elo, I. T., Hempstead, K., Bor, J., & Preston, S. H. (2021). COVID-19 and excess mortality in the United States: A county-level analysis. *PLoS medicine*, 18(5), e1003571.

Weinberger, D. M., Chen, J., Cohen, T., Crawford, F. W., Mostashari, F., Olson, D., ... & Viboud, C. (2020). Estimation of excess deaths associated with the COVID-19 pandemic in the United States, March to May 2020. *JAMA internal medicine*, 180(10), 1336-1344.

Wilson, T., Temple, J., & Charles-Edwards, E. (2022). Will the COVID-19 pandemic affect population ageing in Australia?. *Journal of Population Research*, 39(4), 479-493.

Reviewers' Comments:

Reviewer #1:

Remarks to the Author:

Dear Author(s),

Thank you for adequately addressing all the comments I previously provided. Hence, regarding the points I raised in my review no further actions are required. The author(s)' responsiveness to feedback is greatly appreciated.

Reviewer #2:

Remarks to the Author:

The authors have addressed all my concerns.

Reviewer #4:

Remarks to the Author:

The authors have addressed most of the concerns of the reviewers, also by adding additional figures, further information about data used and alternative data sources available, and generating 95% intervals around their estimates & projections. The data limitations and assumptions are better explained.

I still have a few suggestions that can be addressed in the final revision

Estimating the fertility impact of the pandemic

This is in my view the least plausible component of the exercise, partly due to the annual format of the data used, partly due to attributing the 2020 fertility shift (compared with the "expected" fertility) entirely to the covid pandemic, and partly due to using 2021 fertility data as a baseline estimate of "post-covid" fertility rates. Here the disadvantage of working with annual data becomes most prominent: considering the average length of pregnancy close to 9 months, the covid-19 pandemic would be expected to start affecting births only since October-November 2020, i.e., for the women getting pregnant at the time the pandemic started having a large impact on the US. population in early 2020. In a similar way, most of the births occurring in 2021 were conceived in 2020—clearly still during the pandemic period. This means there is an analytical mismatch between the finer data needed to estimate pandemic effect on fertility and the data and assumptions actually used.

This mismatch could be addressed by moving away from the annual UNWPP data and adopting recent estimates of the COVID-19 impact on US birth trends in 2020 and 2021 by Bailey et al. (2023), which is summarised by year, but based on monthly data (while still using UNWPP age-specific fertility rates, adjusted for the TFRs published in Bailey et al.). The Bailey et al. study also provides a plausible explanation for the earlier fertility decline (before October 2020) in the US, linked to lower fertility among migrant women, likely connected with lower immigration (of pregnant women) during 2020—and thus attributable to immigration restrictions during the pandemic.

Interpreting findings in perspective

At times the authors appear to exaggerate the relevance of the covid-related population shifts they describe. They repeatedly write about "rippling effects" of the pandemic for the US population size and structure (or, at another place, writing about a "massive reduction" of US working age individuals). Seen from a broader perspective, their finding that without the pandemic the US population would be by 0.63% larger in 2025 and 0.44% larger in 2060 does not look so dramatic (and neither do the described changes in the age structure). In comparison, annual immigration rate to Canada has recently (2021 and 2022) amounted to over 1% of Canada's population—modifying the population size and structure more sharply in just one year than was the cumulative estimated impact of the

pandemic on the US population. Economic shocks and migration waves may also have a more sizeable impact on population size and structure—and covid pandemic itself had a larger impact on populations of other countries (as shown in the study on Spain by Gonzales-Leonardo and Spijker 2022). Therefore, I suggest avoiding strong and dramatic terms when commenting the findings, which are relevant, but not extraordinary.

Figures and Tables

Although the study is complemented with many figures, I miss a summary table that would show the estimated US population loss (absolute and relative) due to covid-related mortality, fertility and migration in 2022 or 2025 and 2040 or 2060.

Future research: broadening the focus

This study is focused on the US population trends. Given that the UNWPP datasets used in the analyses are available for all countries in the world, a similar setup could be used to compute corresponding estimates for different countries and regions or, indeed, for all countries globally (assuming the authors would be able to revise their method of estimating covid impact on fertility rates). This research would put the findings for the US into a wider perspective (probably showing that covid had a more dramatic impact on many other populations). (This is a suggestion for future research, not a comment on this study)

Specific and minor comments:

P. 3: "Insights gained from this exercise (...) will help tailor economic and social policies forward." This is a very general statement – could you provide more specific examples of the policy implications?

P. 4: "during 2020 (...) the US total fertility rate was much lower than in the previous years". Looking at a longer-term TFR trend in the US, from 1.89 in 2011 to 1.64 in 2020, the fall by 0.046, from 1.689 in 2019 to 1.643 in 2020 does not look very exceptional.

P. 5: "...migration represents one important remedy against population ageing". Consider a more nuanced statement – given the long-term increase in life expectancy (in the US, until around 2013), migration can moderate or slow-down population ageing, but cannot serve as its "remedy" in the sense of stopping a rise of old-age dependency ratios.

Response to Reviews

Subject: Revision Number 2, (NCOMMS-23-08414) entitled "The Long-term Effects of the COVID-19 Pandemic on U.S. Population Structure"

Dear Reviewers,

Thank you for reviewing our manuscript entitled "The Long-term Effects of the COVID-19 Pandemic on U.S. Population Structure" for a second time.

We appreciate the supportive comments and the critical concerns. On the pages below, we address the remaining comments. Akin to our letter for the first set of reviews, we document our point-by-point response to the reviewers' comments, cross-referencing similar feedback accordingly. We indicate reviewer feedback in text beginning "C," our responses in text beginning "R," and the manuscript changes in response in "quotes". We have indicated changes in the revised manuscript with red font.

We believe that our manuscript is again strengthened because of this review, and we are hopeful that we were able to adequately address the remaining concerns.

We are of course open to any additional suggestions you might have and look forward to receiving your feedback.

Sincerely,
Authors

Reviewer #1 (Remarks to the Author):

C1. Dear Author(s),

Thank you for adequately addressing all the comments I previously provided. Hence, regarding the points I raised in my review no further actions are required. The author(s)' responsiveness to feedback is greatly appreciated.

R1. We are pleased to hear this. Additionally, we appreciate the positive reception of our revisions. Thank you.

Reviewer #2 (Remarks to the Author):

C2. The authors have addressed all my concerns.

R2. Thank you; we appreciate hearing this.

From Reviewer 4, on behalf of Reviewer 3:

[Note from Authors: Reviewer 3 was unavailable, and, at the editor's request, Reviewer 4 provides a few suggestions for how we can address additional concerns pertaining to Reviewer 4's original concerns. We address these below as though we are writing to Reviewer 3.]

C3. The study needs to reflect more extensively the limitations of the assumptions, methods (counterfactual analysis), and data used. For instance, the authors should acknowledge that their findings may not replicate outside of the US.

R3. We agree that discussing limitations is imperative with all analyses, especially those like ours that involve many assumptions. We have added an important clarification about this in our Results section (Page 4, Paragraph 2):

“Although it is unlikely that mortality, fertility, or migration were entirely unaffected by pandemic changes in the respective remaining demographic processes, we attempt to isolate the independent contributions from mortality, fertility, and migration on population structure by running our model as though counterfactual conditions only applied to one demographic process at a time (see Materials and Methods).”

Additionally, as a gentle reminder to the reviewer, we have three paragraphs worth of limitations, spanning Pages 6 and 7 of the manuscript. We have added an additional clarification (see red text below). We paste these paragraphs here for the reviewer's reference:

“Although the UNWPP data represent a gold standard in terms of population projections, our counterfactual analysis is subject to three limitations. First, our findings are based on UNWPP's medium scenario, i.e., not the most aggressive or the most conservative estimate. As the baseline mortality, fertility, and migration rates and counts represent forecasts themselves, they are subject to uncertainty, which is carried over to our counterfactual estimates. We attempt to mitigate this by focusing on the difference between baseline and counterfactual scenarios. Thus, because mortality, fertility, and migration conditions are set to equal after 2024, there is little room for forecasting errors to compound over time, as these will mostly cancel out. Moreover, the published UNWPP forecasts for the year 2022 correspond well with preliminary estimates of mortality, fertility, and migration (4, 6, 18) generating further trust in our baseline and counterfactual estimates for the pandemic period. **Additionally, due to the nature of counterfactual analyses, it is not possible to truly know what observed rates and counts would have been in the absence of the pandemic. While we estimate these to the best of our ability, all analyses must be considered with this limitation in mind.**

Second, our finding that changes in migration during the pandemic exert the biggest long-term effects on population size may partially be driven by the lack of adequate age- and sex-specific migration counts for the U.S. and the application of model migration schedules (38) for both the baseline and the counterfactual scenario. We assume a family migration schedule, with migrants concentrated in young and working

ages. This also means that the second-order effects of migration through never-born children are particularly large in our study. Immigration to the U.S. has traditionally been concentrated in working ages (39) and it is plausible that the largest declines in migration during the pandemic occurred in these age groups. Although it is entirely possible that migration decreased more in other age-groups, including ages older than reproductive ages, existing data on foreign-born immigration to the U.S. indicate that different types of migration (i.e., refugees/asylum seekers, students, work visas, immigrant visas) were similarly affected during the pandemic (6). Moreover, the enactment of Title 42 during the pandemic was likely responsible for much of the decline in migration to the U.S. and targeted a broad range of countries. Thus, our decision to use similar migration schedules for our baseline and counterfactual scenario appears justified. While we are limited by the lack of migration data at smaller temporal windows (e.g., month or week), future work with better data availability might consider analyzing this to gain a more nuanced understanding of how these processes vary across other temporal dimensions.

Third, following UNWPP, we assume that mortality, fertility, and migration return to their pre-pandemic trajectories after a few years. There is inconclusive evidence about what signals the end of a pandemic or epidemic (40), so it is possible that the assumptions from UNWPP are incorrect. Should that be the case, and mortality continue to remain higher than expected, and fertility and/or migration continue to remain lower than expected, then our estimates represent an underestimation. However, the indirect consequences of the pandemic may continue to negatively affect the U.S. mortality, fertility, and migration environments well into the future, and we are not able to measure these indirect consequences here. First, long COVID and unmet healthcare needs during the pandemic may increase the risk of mortality in the long run. Other consequences of the pandemic, such as the loss of next of kin (41), learning loss (42), or racist and xenophobic behavior against Asians and Asian-Americans (43, 44) may also exert negative effects on population health and mortality for generations to come. Second, the experience of economic uncertainty and stress related to the balancing of work and childcare obligations during the pandemic may have raised doubts among some couples about having (additional) children in the future (45, 46). Finally, migration to the U.S. may remain below expected levels in the future, as some individuals who would have migrated to the U.S. may have died during the pandemic, or established families in their country of origin or other countries with less restrictive migration policies. Based on these reflections about the potential long arm of the pandemic, the findings presented in this manuscript, which assume a short pandemic shock, most likely represent a lower bound.”

We are also careful throughout the manuscript and in the manuscript title to reference results as they pertain to the U.S. – a few examples from the conclusion here:

“Our study provides first results on how the pandemic’s reshaping of the U.S. population will repercuss into the future.” (Discussion, First sentence, Page 5)

“Accordingly, our investigation is the first of its kind covering the United States. Because the United States is known for having exceptionally high COVID-19 mortality (52), it is particularly striking to find that pandemic-induced migration changes will have a

comparatively large and longer-lasting effect on population size.” (Discussion, Final 2 sentences, Page 7)

We take the reviewer’s point that we ought to **“acknowledge that [our] findings may not replicate outside of the US.”**, along with the suggestion from Reviewer 4 (C9) that it would be interesting to have results for other countries, and we now add a sentence to Paragraph 8 of the Discussion (Page 7), which reads as:

“It will also be valuable to apply this approach to other countries (beyond Spain and Australia (8, 9)), as the pandemic unequally affected each nation.”

C4. The authors need to be more transparent about the limitations of the method used to estimate uncertainty, namely, that it does not address the underlying uncertainty about the assumptions they made when accounting for the covid impact on births, deaths and migration.

R4. Thanks for pointing out that we hadn’t explicitly acknowledged this. We now add two sentences to our Materials and Methods section (Page 8, Paragraph 3) to explicitly state this:

“Importantly, this method for generating confidence intervals does not address underlying uncertainty around the estimation of birth and death rates as well as migration counts. We refer to our Online Appendix for supplementary analyses with alternate assumptions.”

Reviewer #4 (Remarks to the Author):

C5. The authors have addressed most of the concerns of the reviewers, also by adding additional figures, further information about data used and alternative data sources available, and generating 95% intervals around their estimates & projections. The data limitations and assumptions are better explained.

I still have a few suggestions that can be addressed in the final revision

R5. Thank you for your positive reception of the changes we made in the revision. We hope that we were able to satisfactorily address your final suggestions here.

C6. Estimating the fertility impact of the pandemic.

This is in my view the least plausible component of the exercise, partly due to the annual format of the data used, partly due to attributing the 2020 fertility shift (compared with the “expected” fertility) entirely to the covid pandemic, and partly due to using 2021 fertility data as a baseline estimate of “post-covid” fertility rates. Here the disadvantage of working with annual data becomes most prominent: considering the average length of pregnancy close to 9 months, the covid-19 pandemic would be expected to start affecting births only since October-November 2020, i.e., for the women getting pregnant at the time the pandemic started having a large impact on the US. population in early 2020. In a similar way, most of the births occurring in 2021 were conceived in 2020—clearly still during the pandemic period. This means there is an analytical mismatch between the finer data needed to estimate pandemic effect on fertility and the data and assumptions actually used.

This mismatch could be addressed by moving away from the annual UNWPP data and adopting recent estimates of the COVID-19 impact on US birth trends in 2020 and 2021 by Bailey et al. (2023), which is summarised by year, but based on monthly data (while still using UNWPP age-specific fertility rates, adjusted for the TFRs published in Bailey et al.). The Bailey et al. study also provides a plausible explanation for the earlier fertility decline (before October 2020) in the US, linked to lower fertility among migrant women, likely connected with lower immigration (of pregnant women) during 2020—and thus attributable to immigration restrictions during the pandemic.

R6. These points and suggestions from the reviewer are important. As we understand their concern: the linear interpolation we used for estimating a fertility counterfactual for 2020 is inadequate. This is because we assumed that 2021 rates reflect a continuation of pre-pandemic trends. This is not the case in the U.S., as the Bailey et al. (2023) study show that there was in fact a “baby bump.”

There are many ways to operationalize counterfactuals, and the paper from Bailey et al. (2023) uses a linear projection based on 2015–2019. We considered a similar approach, but we were not entirely satisfied with the reliability. We demonstrate this here.

We compared observed values for the Total Fertility Rate (TFR) between 2010 and 2021 (black dots) with those ‘predicted’ based on a linear trend (a) between 2010–2019 (solid black line); (b) between 2015–2019 (solid red line). We use three different data sources to test the robustness of our comparisons: the United Nations World Population Prospects (UNWPP) (Response Figure 1), the Human Mortality Database (HMD) (Response Figure 2), and the National Vital Statistics System (NVSS) (Response Figure 3).

Response Figure 1. Observed vs. Linear Trend Prediction TFR, UNWPP

Response Figure 2. Observed vs. Linear Trend Prediction TFR, HMD

Response Figure 3. Observed vs. Linear Trend Prediction TFR, NVSS

Based on the TFR values predicted from the 2015–2019 trend, and akin to the results of the Bailey et al. (2023) paper, we would conclude for all three data sets (less so for UNWPP) that there was a *baby bust* in 2020 (observed TFR lower than predicted TFR) and a *baby bump* in 2021 (observed TFR greater than predicted TFR). However, this is not the case if the 2010–2019 trend is used to predict TFR values in 2020 and 2021. In the latter case, we would conclude that the observed TFR has seen a *return* to expected levels in 2021. The Figures illustrate that the conclusion whether there was a baby bump in 2021 hinges on the choice of the starting year for the linear trend: TFR values in 2014–2015 were higher than or similar to the one observed in 2013. This creates a much steeper slope (and a lower predicted TFR value for 2021) if the linear trend is based on 2015–2019 vs. 2010–2019 data.

So, while we take the suggestion from the reviewer in stride and consider amending how we calculated our counterfactual estimate, we ultimately prefer to stick with our initial calculation (the linear interpolation between 2019 and 2021).

Additionally, we appreciate that the reviewer noted the publication of the Bailey et al. (2023) paper in PNAS. We were citing a previous version of the manuscript, that was published at NBER. We have updated our manuscript to now cite the new version of Bailey et al.

C7. Interpreting findings in perspective.

At times the authors appear to exaggerate the relevance of the covid-related population shifts they describe. They repeatedly write about “rippling effects” of the pandemic for the US population size and structure (or, at another place, writing about a “massive reduction” of US working age individuals). Seen from a broader perspective, their finding that without

the pandemic the US population would be by 0.63% larger in 2025 and 0.44% larger in 2060 does not look so dramatic (and neither do the described changes in the age structure). In comparison, annual immigration rate to Canada has recently (2021 and 2022) amounted to over 1% of Canada’s population—modifying the population size and structure more sharply in just one year than was the cumulative estimated impact of the pandemic on the US population. Economic shocks and migration waves may also have a more sizeable impact on population size and structure—and covid pandemic itself had a larger impact on populations of other countries (as shown in the study on Spain by Gonzales-Leonardo and Spijker 2022). Therefore, I suggest avoiding strong and dramatic terms when commenting the findings, which are relevant, but not extraordinary.

R7. We appreciate the reviewer’s encouragement to be honest but not dramatic. This is valuable feedback. We have edited the manuscript in several places to dampen this language and list these places below. We do, however, advocate for still referring to effects as “rippling,” because this indicates that the effects carry on into other domains or into the future. As the reviewer will note below, we remove amplifying words around “ripple” though.

- Page 3: deletion of the sentence “Insights gained [...] social policies forward” (See C10 (A) below for more context).
- Page 3: addition of the word “slightly” (“When this is combined with slightly lower fertility rates [...]”)
- Page 3: deletion of the word “persistently” (“there will likely be fewer reproductive-aged [...]”)
- Page 3: replace word “dominate” with “explain” (“ [...] increases in mortality explain the pandemic’s long-term effects [...]”)
- Page 4: removal of word “massive” (“The halting of in-migration under Title 42 likely contributed to a reduction in working-aged individuals”)
- Page 4: replacement of words “will likely” with “may” (“ [...] this is a cycle that may carry on for many years”)
- Page 5: deletion of the word “dramatic” (“Our results show that the decline [...]”)
- Page 6: deletion of the clause “an effect that will likely persist for years to come”
- Page 7: removal of the word “far” (“They ripple beyond immediate, independent changes to mortality, fertility, and migration [...]”)

C8. Figures and Tables

Although the study is complemented with many figures, I miss a summary table that would show the estimated US population loss (absolute and relative) due to covid-related mortality, fertility and migration in 2022 or 2025 and 2040 or 2060.

R8. We understand the desire for a summary table. As a reminder to the reviewer, we already made exactly this suggestion. It is available in our Appendix (Section V, Supplementary Table 2, Page 16). While we have a preference to keep this table in the appendix, we can move it to the main text if the reviewer and editor require.

C9. Future research: broadening the focus

This study is focused on the US population trends. Given that the UNWPP datasets used in the analyses are available for all countries in the world, a similar setup could be used to compute corresponding estimates for different countries and regions or, indeed, for all countries globally (assuming the authors would be able to revise their method of estimating covid impact on fertility rates). This research would put the findings for the US into a wider perspective (probably showing that covid had a more dramatic impact on many other populations). (This is a suggestion for future research, not a comment on this study)

R9. We agree that this would be interesting! To acknowledge this, and to address Reviewer 3's comment (C3), we add a sentence that reads:

“It will also valuable to apply this approach to other countries (beyond Spain and Australia (8, 9)), as the pandemic unequally affected each nation.” (Discussion, Paragraph 8, Page 7)

C10. Specific and minor comments:

- A. P. 3: “Insights gained from this exercise (...) will help tailor economic and social policies forward.” This is a very general statement – could you provide more specific examples of the policy implications?**
- B. P. 4: “during 2020 (...) the US total fertility rate was much lower than in the previous years”. Looking at a longer-term TFR trend in the US, from 1.89 in 2011 to 1.64 in 2020, the fall by 0.046, from 1.689 in 2019 to 1.643 in 2020 does not look very exceptional.**
- C. P. 5: “...migration represents one important remedy against population ageing”. Consider a more nuanced statement – given the long-term increase in life expectancy (in the US, until around 2013), migration can moderate or slow-down population ageing, but cannot serve as its “remedy” in the sense of stopping a rise of old-age dependency ratios.**

R10.

- A. We agree, this was a quite general statement. Additionally, in line with the reviewer comment above (C7), we are worried this was a bit of an oversell. As such, we have deleted the sentence entirely.
- B. This is a fair point. We've dampened the language here by removing the word “much.”
- C. Thanks for drawing our attention to this problematic language. We've rewritten the sentence, it now reads as: “[...] migration represents one important **mechanism for slowing down** population aging.”

Reviewers' Comments:

Reviewer #4:

Remarks to the Author:

The authors have adequately addressed my concerns. I have no additional comments or suggestions.